# Early patterning and specification of cardiac progenitors in gastrulating mesoderm

W Patrick Devine[1,2,3,5]*, Joshua D Wythe[1,2], Matthew George[1,2,4], Kazuko Koshiba-Takeuchi[1,2], Benoit G Bruneau[1,2,3,4]*

[1]Gladstone Institute of Cardiovascular Disease, San Francisco, United States; [2]Roddenberry Center for Stem Cell Biology and Medicine at Gladstone, San Francisco, United States; [3]Cardiovascular Research Institute, University of California, San Francisco, San Francisco, United States; [4]Developmental and Stem Cell Biology Program, University of San Francisco, San Francisco, United States; [5]Department of Pathology, University of California, San Francisco, San Francisco, United States

**Abstract** Mammalian heart development requires precise allocation of cardiac progenitors. The existence of a multipotent progenitor for all anatomic and cellular components of the heart has been predicted but its identity and contribution to the two cardiac progenitor 'fields' has remained undefined. Here we show, using clonal genetic fate mapping, that *Mesp1+* cells in gastrulating mesoderm are rapidly specified into committed cardiac precursors fated for distinct anatomic regions of the heart. We identify *Smarcd3* as a marker of early specified cardiac precursors and identify within these precursors a compartment boundary at the future junction of the left and right ventricles that arises prior to morphogenesis. Our studies define the timing and hierarchy of cardiac progenitor specification and demonstrate that the cellular and anatomical fate of mesoderm-derived cardiac cells is specified very early. These findings will be important to understand the basis of congenital heart defects and to derive cardiac regeneration strategies.

*For correspondence: patrick. devine@gladstone.ucsf.edu (WPD); bbruneau@gladstone. ucsf.edu (BGB)

**Competing interests:** The authors declare that no competing interests exist.

**Reviewing editor**: Marianne E Bronner, California Institute of Technology, United States

## Introduction

Mammalian heart development involves the allocation of cardiac progenitors in a discrete spatial and temporal order (*Evans et al., 2010*; *Bruneau, 2013*). Understanding the identity and regulation of these progenitors is critical to understanding the origins of congenital heart defects and may lead to novel cell-based regenerative therapies for heart disease (*Bruneau, 2008*; *Xin et al., 2013*). The existence of an early and specific multipotent progenitor for all anatomic and cellular components of the heart has been predicted (*Parameswaran and Tam 1995*; *Tam et al., 1997*; *Meilhac et al., 2004b*; *Kinder et al., 2001*), but the identity of this progenitor and when it arises in embryonic development has remained undefined. At least two sets of molecularly and morphologically distinct cardiac precursors have been identified in the mammalian embryo, referred to as the first and second heart fields; these populations contribute to distinct anatomical structures within the heart (*Evans et al., 2010*; *Buckingham et al., 2005*). Separation of the left and right ventricles is dependent on a single structure, the interventricular septum (IVS) and it has been postulated that the IVS myocardium has a dual contribution from these two heart fields (*Bruneau et al., 1999*; *Takeuchi et al., 2003*). The existence of these two cardiac progenitor 'fields' raises the question of when cardiac precursors are allocated to these populations and their contributions to mature structures in the heart, such as the IVS.

**eLife digest** Most internal organs in the body are made up of several different kinds of cells. Understanding where these cells come from and how these different cells develop from a single cell in an embryo could help to guide regenerative therapies, where tissues grown in the laboratory are used to repair damage that the body cannot repair itself.

The existence of a single heart progenitor cell that can produce all of the heart's structures has long been predicted, but has so far escaped discovery. Currently, it is known that two distinct sets of heart precursor cells exist in mammals, which each produce cells for different parts of the heart. Work performed in mouse embryos has hinted that both sets of cells develop from cells that produce a protein called Mesp1. This protein controls when many genes—including those involved in heart development—are activated.

Devine et al. marked a small number of Mesp1-producing cells and followed the fate of these cells through development to see where their descendants would end up within the embryo—and specifically within the mature heart. Labeling occurred at a very early stage of development, called gastrulation, when the embryonic cells first begin to organize themselves into three tissue layers that will go on to form all the different parts of the organism. Devine et al. found that shortly after gastrulation begins, heart precursor cells are present and are already assigned to particular regions of the heart. This means that if there is a single pool of heart precursor cells, it specializes into different populations very early in the development of an embryo.

Devine et al. show that during gastrulation, heart precursor cells are already split into two distinct populations: one containing the cells that go on to form the atria and left ventricle of the heart; the other consisting of the cells that will make up the right ventricle and the 'outflow tract' that will eventually form the great vessels leading into and out of the heart. These two populations are separated by a boundary, which Devine et al. suggest is established very early on, and will go on to form the septum that separates the left and right ventricles in the developed heart. As defects in the septum are the source of many congenital heart defects, a better understanding of the heart cell precursor populations and how they interact could help develop treatments for these conditions.

Previous studies have suggested that a population of early mesoderm expressing the transcription factor Mesp1 precedes the establishment of the anatomically and molecularly distinct 'heart fields' (*Saga et al., 1996*, *1999*; *Bondue et al., 2008*; *Lindsley et al., 2008*) that ultimately will differentially populate the great vessels, RV and atria or the LV and atria. However it is clear that *Mesp1+* cells can also contribute to a broad range of mesodermal derivatives that include, but are not restricted to, the developing heart (*Chan et al., 2013*; *Yoshida et al., 2008*). A population of mesoderm labeled by *Eomesodermin* has also been shown to contribute to the developing heart, but again these cells have broad contributions in the embryo (*Arnold and Robertson, 2009*). Retrospective lineage analysis supports the distinct origins of segments of the heart from individual precursor pools (*Meilhac et al., 2003*; *Buckingham et al., 2005*; *Meilhac et al., 2004b*), but several questions remain regarding the timing and molecular progression of cardiac specification (*Meilhac et al., 2004b*). For example, do early mesodermal cells become 'locked into' a cardiac fate early on and when do they become 'assigned' to an anatomical location? Is there a multipotent, specified cardiac progenitor that anticipates the currently understood heart fields?

Here we show that early cardiac progenitors are assigned to a specific developmental path prior to or shortly after the initiation of gastrulation. We identify a population of specified cardiac precursors arising from these mesodermal progenitors that express the chromatin remodeling factor *Smarcd3* prior to the onset of expression of known cardiac progenitor markers (*Nkx2-5*, *Isl1*, and *Tbx5)*. Clonal labeling of early cardiac precursors highlights the heterogeneity among Mesp1+ progenitors, including the existence of precursors that are restricted in their anatomical contribution, especially to distinct ventricular chambers. Finally, inducible genetic marking of early *Tbx5+* and *Mef2c-AHF +* populations highlights this early segregation of cardiac progenitors and suggests that the compartment boundary that exists between the right and left ventricles arises from an early clonal boundary, prior to the onset of septum morphogenesis. Overall our findings delineate the progression and molecular identity of cardiac precursors in the early mouse embryo.

# Results

In reassessing the in vivo differentiation potential of Mesp1+ cells, we find that this population contributes broadly to several mesodermal derivatives, (*Figure 1A*), consistent with other reports (*Yoshida et al., 2008*). We reasoned that among this diverse mesodermal population, a more specific population destined for the cardiac lineage exists. To test this model, we performed in vivo clonal analysis by generating mosaic mice in which very few *Mesp1*+ cells were labeled at isolated clonal density via the mosaic analysis with double markers (MADM) system (*Zong et al., 2005*; *Hippenmeyer et al., 2010*) (*Figure 1B–C*). This approach is particularly advantageous because labeling events are rare, labeling is permanent, and one can identify labeled daughter cells (twin spots) based on color (*Figure 1—figure supplement 1A*). We analyzed in fetuses (E12.5-E14.5) the anatomic distribution and cellular constituents of clones induced by *Mesp1*$^{Cre}$ (which is active in mesoderm from ~E6.0 to E7.5) (*Saga et al., 1999*). While we did not use a conditional Cre allele to control the timing of Cre activity, we confirmed the timing of *Cre* expression by in situ hybridization (*Figure 1—figure supplement 1B*). By the late head fold stage (LHF), we see a downregulation of *Cre* mRNA and localization to the base of the allantois. We see no expression in the area of forming cardiogenic mesoderm. In addition, we counted the number of labeling events in embryos at E8.5 and E14.5 (*Figure 1—figure supplement 1D–E* and Statistical Analysis) and saw no change in the distribution of labeled clusters, suggesting that no additional recombination events have occurred over this time interval. Finally, a complementary lineage labeling approach using a *Mesp1-rtTA* transgenic allele (*Lescroart et al., 2014*) defines a functional window of Mesp1 activity based on the timing of doxycycline administration between E6.25-E7.5, again supporting the narrow timing of *Mesp1* activity.

In order to ensure an accurate description of clone locations throughout the embryo, a thorough external examination of embryos was performed followed by removal and, in many instances, immunostaining of dissected hearts for labeled twin spots. Coherent clusters of uniquely colored cells separated by > 100 μm were classified as a twin spot derived from a single labeling event. Because the majority of hearts (32 of 38 embryos) contained three or fewer uniquely colored clusters, determining lineage relationships between and among clusters was straightforward. A subset of clones uniquely label the heart (*Figure 1D–E* and *Figure 2*), demonstrating the existence of an early, cardiac-specific progenitor. We also found specimens with clones of cells in other mesodermal derivatives but with no apparent clones within the heart (*Figure 1—figure supplement 1F–G* and *Figure 2*), conclusively demonstrating that within the population of *Mesp1*+ mesoderm a dedicated population of cardiac progenitors exists.

To determine if these early cardiac progenitors represent a common precursor for the two heart fields, we analyzed the anatomic distribution of *Mesp1*$^{Cre}$-MADM twin spot clones within the heart. In contrast to previous retrospective clonal analysis (*Meilhac et al., 2004b*), we did not observe twin spots that populated anatomic structures classically thought to derive from first and second heart field progenitors, for example spanning the left and right ventricles, or contributing to the left ventricle and outflow tract. Rather, we saw twin spots that populated discreet anatomic locations including the left ventricle (*Figure 1F–G* and *Figure 1—figure supplement 3A–B*), right ventricle (*Figure 1D* and *Figure 1—figure supplement 3C–D*), outflow tract (*Figure 1N–O* and *Figure 1—figure supplement 3E–F*), atria-left ventricle (*Figure 1L–M*), and interventricular septum (*Figure 1H*). Notably absent from our clonal analysis were twin spots that spanned the right and left ventricles. Based on the total number of clonal observations made (n = 96), the probability that a common progenitor for the left and right ventricles does not exist, assuming a binomial distribution (success, or 1, = the progenitor is observed; failure, or 0 = the progenitor is not observed), can be calculated using Jeffreys interval. The upper and lower limits of a 95% confidence interval was calculated such that the upper limit is 0.019 (See Statistical Methods). Thus, we can be quite confident given the number of observations made that such a common progenitor does not exist within the *Mesp1*+ population. An explanation for the discrepancy between our *Mesp1*$^{Cre}$-MADM clonal analysis and prior retrospective clonal analyses lies with the timing of the labeling events. *Mesp1* is transiently expressed ((*Saga et al., 1999*) and *Figure 1—figure supplement 1B*), thus all of our clones are induced over a narrow window of time during gastrulation. The retrospective clonal analysis employed a cardiomyocyte-specific promoter, so an early labeling event (in the epiblast for example) that would contribute broadly throughout the embryo would only be visualized in the heart, giving the impression of a specific common progenitor cell. Indeed, when we perform MADM clonal analysis using the epiblast-specific Cre line *Sox2::Cre*, we

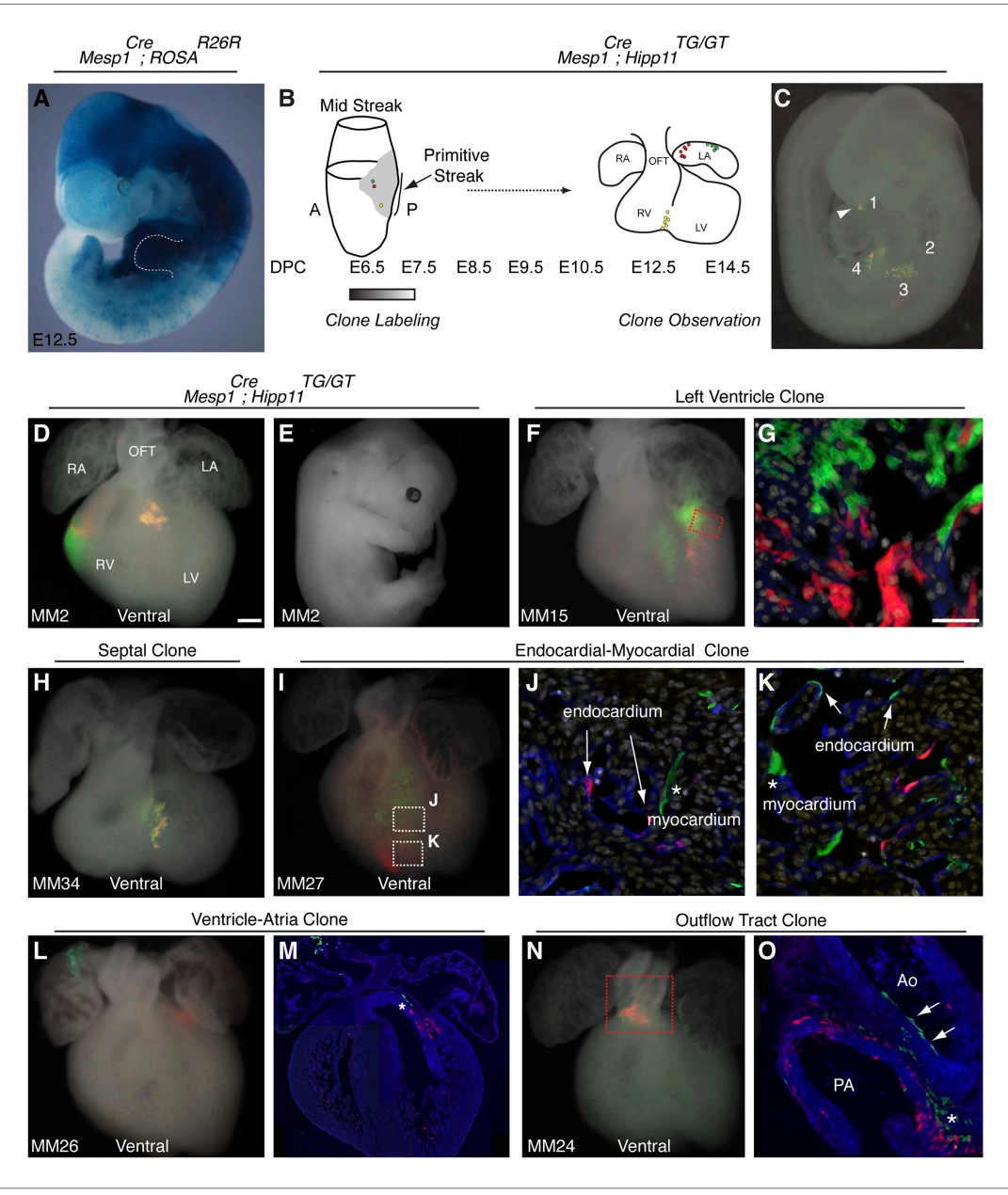

**Figure 1**. The first and second heart fields diverge early in gastrulating mesoderm. (**A**) Genetic lineage tracing of *Mesp1*<sup>Cre</sup>;ROSA<sup>R26R</sup> mice reveals widespread labeling of mesodermal derivatives at E10.5, including forelimb (dotted outline). (**B**) Schematic of experimental protocol. Single cells are labeled early in gastrulating mesoderm and progeny of labeled cells observed later in development. (**C**) Example of clonal labeling in E9.5 embryo. Four distinct, scattered, clusters of labeled cells are present throughout trunk and neck, including a single yellow clone in ventricle (1, arrowhead). (**D**) Ventral view of a second heart field progenitor clone (red and green twin spots) with an additional yellow clone in the septal region (embryo ID MM2). (**E**) Whole mount view revealing an absence of non-cardiac clones elsewhere in the same embryo. (**F**) Ventral view of left ventricle clone (embryo ID MM15). Red and green twin spots are adjacent to each other. (**G**) Section through red-boxed area of embryo MM15 showing intermingling of red and green twin spots. (**H**) Ventral view of large, yellow septal clone (embryo ID MM34). (**I**) Whole-mount ventral view of red and green twin-spots in right ventricle (embryo ID MM27). Boxed regions indicate areas shown in higher magnification sections. (**J**) Section through clone in embryo MM27 reveals green labeled cardiomyocyte (asterisk) and red endocardial twin spot (arrows). Note overlap of red clonal labeling with blue PECAM staining. (**K**) Additional section through clone in embryo MM27. Green twin spot contributes to both cardiomyocytes (asterisk) as well as PECAM stained endocardial cells (arrows). (**L**) Whole-mount ventral view
*Figure 1. Continued on next page*

*Figure 1. Continued*

and (**M**) section of heart at E14.5 with a left ventricle-atria clone (embryo ID MM26). Note red and green twin spots (in LV and RA) in whole-mount view. Sectioning reveals a subset of the green twin spot has remained in the top of the left ventricle (asterisk). (**N**) Whole-mount ventral view of red and green twin spots in out-flow tract from embryo MM24. (**O**) Section through outflow-tract region reveals red twin spot contributing predominantly to pulmonary artery. Green twin spot contributes to both pulmonary artery and aorta. In addition, green twin spot appears to contribute to both endothelial lining of aorta (arrows) as well as cardiomyocytes (asterisk) at base of aorta. In (**G**, **J**–**K**) white: DAPI stained nuclei. In (**J**–**K**) blue: PECAM stained endothelial cells. In (**M** and **O**) blue: phalloidin stained actin. A, anterior; LA, left atrium; LV, left ventricle; OFT, out-flow tract; P, posterior; RA, right atrium; RV, right ventricle. Scale bars: (**G** and **J**–**K**), 100 µm (**D**, **F**, **H**, **I**, **L**, **N**), 200 µm.

The following figure supplements are available for figure 1:

**Figure supplement 1**. Overview of MADM clonal analysis and twin spot labeling.

**Figure supplement 2**. Epiblast-specific induction of MADM clones.

**Figure supplement 3**. Additional examples of early cardiac progenitor clones.

see many clones in the embryos, including large, dispersed clones throughout the looping heart tube (*Figure 1—figure supplement 2*) that could be interpreted as a labeling event in a single, common cardiac progenitor. We cannot definitively conclude, however, that these labeled cells within the heart are all clonally related because of the large number of labeling events outside of the heart. In summary, our results suggest that soon after a heart field progenitor is specified, shortly after the initiation of gastrulation, the anatomic destiny of daughter cells quickly diverge, especially those destined to occupy /contribute to the left or right ventricle, and these fates remain fixed.

In analyzing the cellular composition of *Mesp1^{Cre}*-MADM clones, we noticed that while most twin spots give rise to homogenous cellular progeny (cardiomyocytes or endothelial cells), individual cardiac progenitors (<5%, see *Figure 2*) can contribute to multiple cell types, for example myocardium and endocardium (*Figure 1J–K* and *Figure 1—figure supplement 3J–L*), or endothelium and smooth muscle (*Figure 1O*). While a multipotent progenitor had been predicted from in vitro ES cell-based differentiation models (*Wu et al., 2006*; *Moretti et al., 2006*) this is the first evidence of the existence of such a multipotent progenitor in vivo.

The existence of a dedicated population of cardiac progenitors within gastrulating mesoderm suggests that these cells may have a unique molecular signature. To define the order and timing of early cardiac gene expression in vivo, we examined by in situ hybridization and by ß-galactosidase reporter activity early markers of cardiac differentiation (*Mef2c, Tbx5, Isl1, Mesp1, and Smarcd3*) at the late streak (LS) and early head fold (EHF) stages. We find that *Smarcd3* expression precedes *Isl1* and *Tbx5* in a domain that lies at the anterior-proximal region of the embryo and extends into the extraembryonic tissues (*Figure 3B,D–E*). This expression domain appears coincident temporally but non-overlapping with *Mesp1* expression (*Figure 3A–B* and *Figure 3—figure supplement 1*), but is within the *Mesp1^{Cre}*-derived lineage (*Figure 4O–P*). Activity of the *Mef2cAHF* enhancer (*Verzi et al., 2005*) is also detected at the LS stage (*Figure 3C*) prior to the expression of *Tbx5* and *Isl1*. Previous studies have shown that activity of this enhancer is dependent on *Isl1* (*Dodou et al., 2004*). In order to confirm an absence of *Isl1* expression at the LS stage, we used a reporter allele where a nuclear lacZ (*Isl1^{nLacZ}*) has replaced a short segment of the coding sequence, including the endogenous start codon (*Sun et al., 2007*). While there was robust ß-galactosidase reporter activity at the cardiac crescent stage, no detectable staining was seen in LS stage embryos (*Figure 3—figure supplement 2A–B*), suggesting that initiation of *Mef2cAHF* enhancer expression precedes *Isl1* expression and its initiation may be independent of *Isl1*. Several hours later (EHF stage), weak *Isl1* and *Tbx5* expression is detectable, *Mef2cAHF* activity remains, and *Smarcd3* expression restricts to the embryo proper (*Figure 3G–J*). The VEGF-A receptor Flk1 labels multipotent progenitors that can differentiate into hematopoietic, endothelial, smooth muscle and cardiac lineages (*Ishitobi et al., 2011*; *Kattman et al., 2006*), thus we looked at co-localization of *Mesp1*, Flk1, and *Smarcd3* in late streak embryos. While we find minimal overlap between *Mesp1* and Flk1 and *Mesp1* and a reporter of early *Smarcd3* expression (*Figure 3—figure supplement 1A,C–D*), there is significant overlap of Flk1 with this same reporter of early *Smarcd3 expression* (see below) in

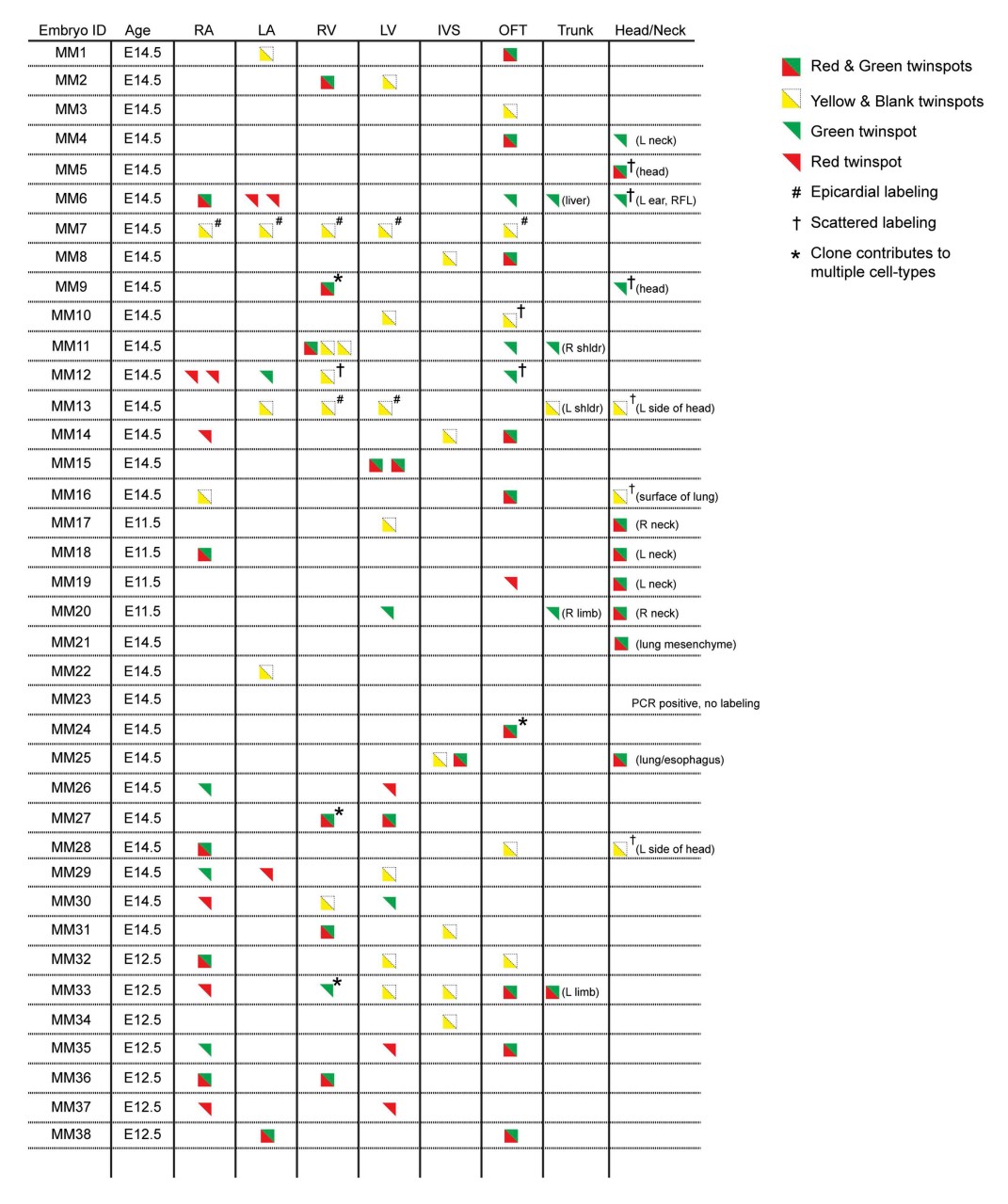

**Figure 2**. Complete description of all *Mesp1^Cre*-MADM clones examined. All observed clones are detailed here, including cardiac as well as extra-cardiac clones. An exhaustive description of extra-cardiac clones is beyond the scope of the current study and thus only a simple description of the tissue or organ containing a labeled clone is included. Red triangles correspond to red twin spots, green triangles correspond to green twin spots, and yellow triangles correspond to yellow twin spots.

late streak stage embryos. Taken together, we see an orderly progression of gene expression that follows the progressive commitment of nascent mesoderm, from early expression of *Mesp1* to the intermediate expression of *Smarcd3*, Flk1, and the *Mef2cAHF enhancer,* and finally the later cardiac lineage-specific expression of *Tbx5* and *Isl1* (**Figure 3K**).

In comparing human and mouse sequences at the *Smarcd3* locus, we identified several regions of conservation in noncoding sequences upstream of the *Smarcd3* transcriptional start site. We tested an ~9 kb genomic element we called *Smarcd3*-F1 in a transgenic mouse reporter assay and found that it was sufficient to recapitulate early endogenous *Smarcd3* expression (**Figure 4A–C**). Expression was

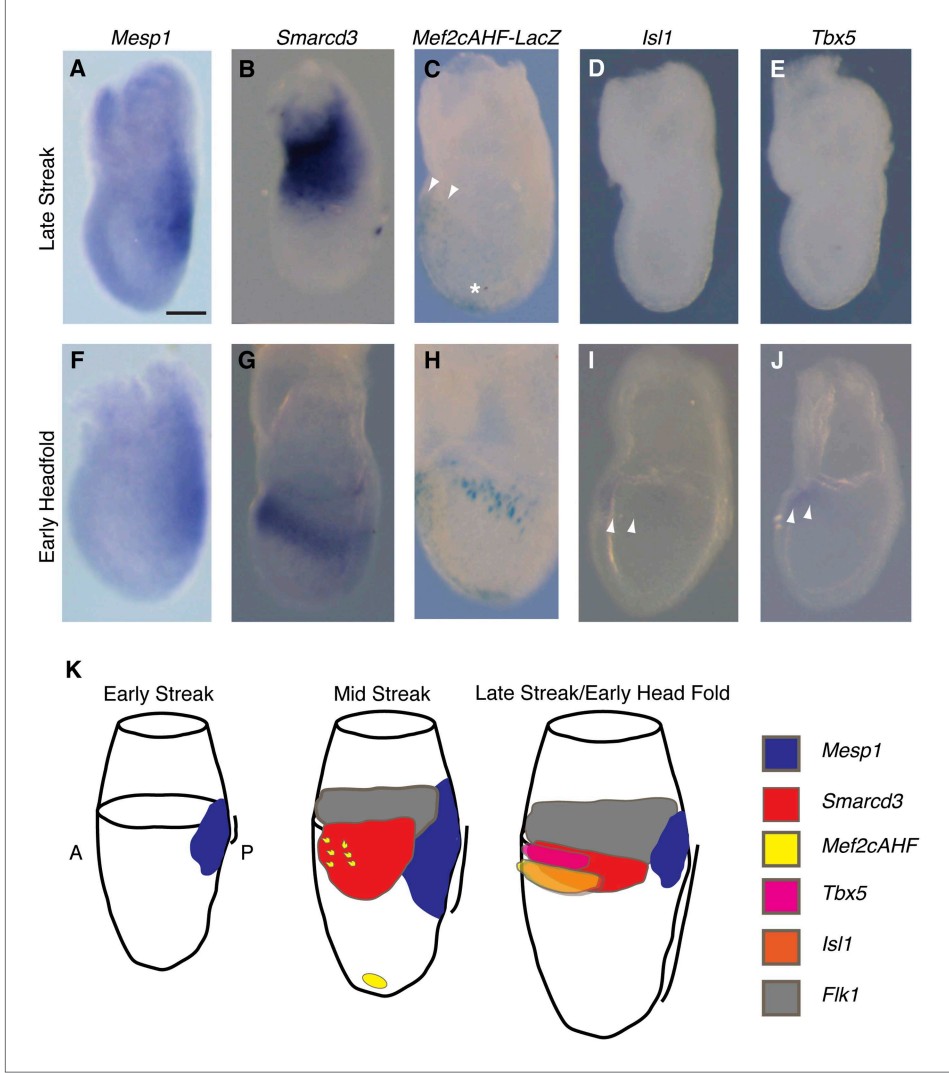

**Figure 3**. Smarcd3 expression initiates in gastrulating mesoderm and precedes expression of *Isl1* and *Tbx5*. (**A–B** and **D–E**) *In situ* hybridization at late-streak (LS) stage for *Mesp1, Smarcd3, Isl1, and Tbx5*. (**C**) X-gal staining at late-streak stage for *Mef2cAHF*-lacZ. *Smarcd3* mRNA is expressed anterior to *Mesp1* mRNA in embryonic and extraembryonic tissues. *Tbx5* and *Isl1* are undetectable by in situ hybridization at this stage. Activity of the *Mef2cAHF* enhancer is detectable around the node (asterisk) and in the anterior embryonic tissues (white arrowheads). (**F–G** and **I–J**) *In situ* hybridization at early-head-fold (EHF) stage for *Mesp1, Smarcd3, Isl1, and Tbx5*. (**H**) X-gal staining at early-head-fold (EHF) stage for *Mef2cAHF*-lacZ enhancer. *Isl1* and *Tbx5* expression is now detectable (arrowheads). (**K**) Summary of gene expression. *Mesp1* expression (blue) precedes all other genes. *Flk1* (gray), *Smarcd3* (red), and *Mef2cAHF* (yellow) expression follows, beginning at the mid-streak stage, in overlapping domains. *Isl1* (orange) and *Tbx5* (magenta) expression begins at the late-streak/early head fold stage and overlaps with the stripe of *Smarcd3* expression. *Mesp1* expression at this stage is restricted to a small domain at the posterior of the embryo. Scale bars: (**A–J**) (100 μm).

The following figure supplements are available for figure 3:

**Figure supplement 1**. Additional characterization of *Smarcd3*-F1-LacZ expression.

**Figure supplement 2**. Characterization of early Isl1[nlacZ] expression.

maintained in the cardiac crescent and looping heart at later stages of development (**Figure 4E–I**), however expression in extraembryonic tissues was noted. Given the broad expression domain of *Smarcd3* mRNA and our *Smarcd3*-F1::*lacZ* reporter lines, we sought to define an enhancer fragment

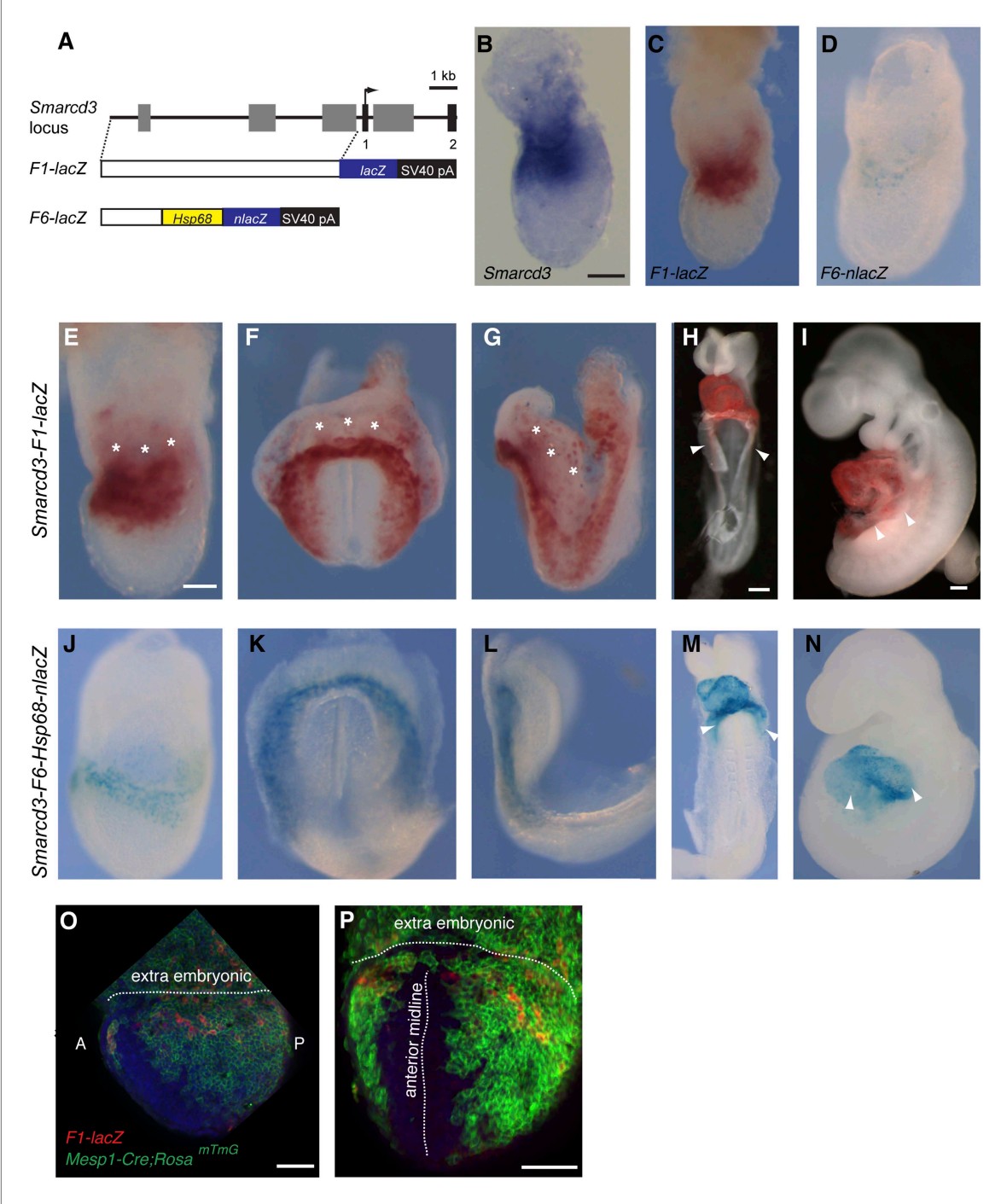

**Figure 4**. Identification of an early cardiac specific enhancer of *Smarcd3*. (**A**) Genomic region upstream of *Smarcd3* translational start site (black arrow). Grey boxes: regions of non-coding sequence conservation between human and mouse. Indicated regions were used to generate *Smarcd3*-F1-lacZ and *Smarcd3*-F6-nlacZ alleles. (**B–C**) Salmon-gal staining of *Smarcd3*-F1-lacZ allele (red) closely mimics endogenous expression of *Smarcd3* mRNA (dark blue). (**D**) X-gal staining of *Smarcd3*-F6-nlacZ allele (light blue) labels a fraction of total *Smarcd3* mRNA. (**E–I**) lateral and frontal views of salmon-gal stained *Smarcd3*-F1-LacZ embryos at (**E**) early head fold (EHF), (**F–G**) cardiac crescent, (**H**) E8.5, and (**I**) E9.5 stages. Note the staining in extraembryonic tissues at EHF and cardiac crescent stages (asterisks). Also note salmon-gal staining in lateral mesoderm of E8.5 and E9.5 embryos (arrowheads). (**J–N**) A single copy of the F6 enhancer along with Hsp68 minimal promoter and nls-LacZ coding sequence were targeted to the *Hipp11* locus on chromosome 11 (see extended methods for details). Lateral and frontal views of X-gal stained *Smarcd3*-F6-Hsp68-nLacZ embryos at (**J**) early head fold (EHF), (**K–L**) cardiac crescent, (**M**) E8.5, and (**N**) E9.5 stages. Note absence of staining in extraembryonic tissues at EHF and cardiac crescent stages as well as restricted cardiac expression at E8.5 and E9.5 (arrowheads). (**O**) Lateral view of late streak stage *Mesp1^Cre*; ROSA^mTmG; *Smarcd3*-F1-lacZ embryo

*Figure 4. Continued on next page*

*Figure 4. Continued*

showing partial overlap of *Smarcd3* expression with the Mesp1-derived lineage. (**P**) Additional anterior view. Blue: DAPI stained nuclei, Green: GFP staining, Red: Beta-galactosidase. Scale bars: (**B**–**D**, **E**–**G**, **J**–**L**), 100 µm, (**H** and **M**), 100 µm, (**I** and **N**), 100 µm.
The following figure supplement is available for figure 4:

**Figure supplement 1**. Generation of *Smarcd3*-F6nLacZ reporter mice.

that might uniquely label early cardiac progenitors. Through deletion analysis of the 9 kb enhancer/promoter region (*Smarcd3*-F1), we defined an ~2.5 kb genomic element (*Smarcd3*-F6) that is expressed only in the embryo proper, in a more restricted pattern than *Smarcd3*-F1 (***Figure 4D and J–N*** and ***Figure 4—figure supplement 1***).

Based on the temporal and spatial expression of *Smarcd3*, prior to expression of the canonical heart field markers *Tbx5* and *Isl1*, and within a small subset of *Mesp1^Cre^*-derived cells, we hypothesized that *Smarcd3*+ cells may represent cardiac precursors. To define the lineage potential of *Smarcd3*+ cells and to compare the lineage potential of the cell populations marked by our two enhancer elements, we performed temporally regulated genetic fate-mapping using mice expressing a tamoxifen inducible CreERT2 under control of the *Smarcd3*-F1 and *Smarcd3*-F6 sequences (*Smarcd3*-F1CreERT2 and *Smarcd3*-F6CreERT2 mice; ***Figure 5—figure supplement 1A–H***). Lineage labeling of *Smarcd3*-F1CreERT2; *Rosa^R26R^* at E5.5 and observation at E10.5 marked cells that contribute predominantly to the heart and anterior forelimb (***Figure 5A–B***), with scattered cells in the trunk and cranial mesoderm (***Figure 5A*** and data not shown); observation at earlier time points revealed significant labeling in extraembryonic tissues (***Figure 5—figure supplement 2A–C***). Within the heart, labeled cells contribute to cardiomyocyte, endocardial, and pericardial layers (***Figure 5C***). Labeling one day earlier (E4.5) marks a similar distribution of cells but with reduced labeling (***Figure 5—figure supplement 2D–F***). Earlier induction (E3.5) and non-injected animals have minimal to no detectable labeling (***Figure 5—figure supplement 2G–K*** and data not shown). Given the restricted expression pattern of the *Smarcd*-F6 enhancer, we hypothesized that marking these cells early on would label a more restricted population of cells later in the embryo. Indeed, lineage tracing with *Smarcd3*-F6CreERT2 labeled a much smaller number of more spatially defined cells, including the heart and a small group of cells in the anterior forelimb (***Figure 5D–E***). There was minimal to no labeling in head/trunk mesoderm or in extraembryonic tissues, and within the heart *Smarcd3*-F6CreERT2 labeled primarily the myocardial layer (***Figure 5F*** and ***Figure 5—figure supplement 3A–C***), consistent with this enhancer labeling a more restricted population of cells. We addressed the clonal potential of early *Smarcd3*+ cells with limiting doses of tamoxifen to induce small numbers of well-isolated labeled cells in *Smarcd3*-F1CreERT2;*Rosa^R26R^* embryos (***Figure 5G–H***). The shape and distribution of the labeled patches is reminiscent of previous work describing the oriented clonal cell growth throughout the myocardium (***Meilhac et al., 2004a***). In order to confirm the clonal nature of these patches induced with our *Smarcd3*-F1CreERT2 allele, we attempted to perform MADM analysis with this particular allele. Unfortunately, the recombination frequency was too low to identify any clones in several litters of embryos and we proceeded with using the *Rosa^Confetti^* multicolor reporter (***Snippert et al., 2010***). We identified 6 *Smarcd3*-F1CreERT2;*Rosa^Confetti^* embryos from several litters where tamoxifen was administered at E5.5, which contained rare or infrequent labeling events; none of these examples showed a labeling pattern that would support a common progenitor contributing to both right and left ventricles. Instead, we saw isolated, single-color clones (***Figure 5I*** and ***Figure 5—figure supplement 3D***) that contributed to a single chamber. While the number of embryos we examined is not sufficient to reach statistical significance, the results are highly supportive of our conclusions from the Mesp1Cre-MADM analysis. We conclude that *Smarcd3* expression in the late gastrulating embryo labels a defined population of specified cardiac precursors that are fated to occupy unique anatomic structures within the mature heart.

Given the restricted expression and lineage potential of cells marked by the *Smarcd3*-F6 enhancer, we sought to characterize the global molecular signature that uniquely identifies this population. We generated a mouse ES cell line with a targeted insertion of the *Smarcd3*-F6 enhancer driving expression of a fluorescent reporter and we differentiated these cells into cardiomyocytes using an established directed differentiation protocol (***Figure 6A*** and ***Figure 6—figure supplement 1***) (***Wamstad et al., 2012***). In this system, we find expression of *Mesp1* is rapidly induced at early mesoderm stage

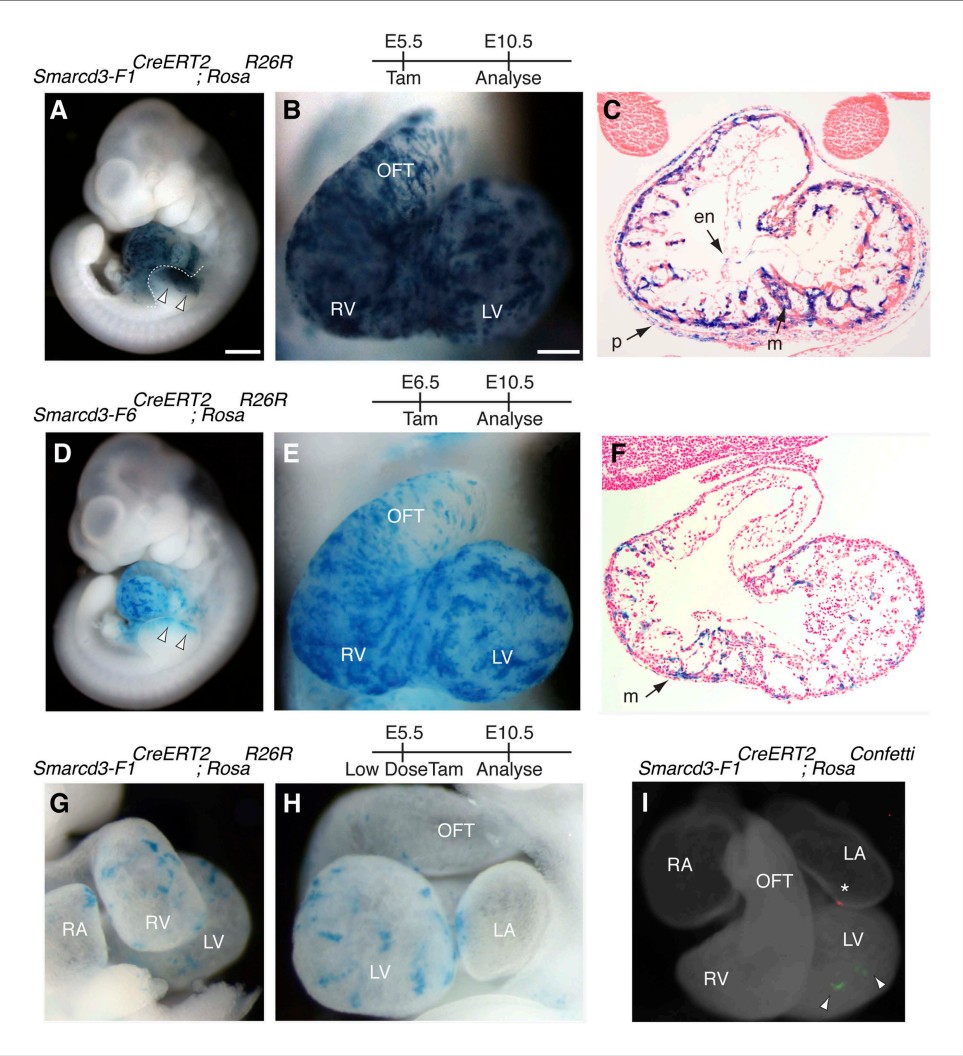

**Figure 5**. A *Smarcd3* enhancer in the late gastrulating embryo labels a population of specified cardiac precursors. (**A**) *Smarcd3*-F1+ cells labeled at E5.5 and observed at E10.5 contribute to the heart and anterior forelimb (arrows). In addition, scattered cells are observed in the trunk and neck (not shown). (**B**) Labeled cells are present in all chambers of the heart, including the RV, LV, OFT, and RA and LA (not shown). (**C**) Within the heart, labeled cells contribute to the pericardial layer as well as the cardiomocyte and endocardial cell layers. (**D**) *Smarcd3*-F6+ cells labeled at E6.5 and observed at E10.5 contribute to the heart and anterior forelimb. No additional labeling in the trunk or neck is observed. (**E**) Labeled cells are also present in all chambers of the heart. The number of labeled cells, however, appears reduced. (**F**) Within the heart, myocardial and pericardial (not shown) cells are labeled. (**G**–**H**) Limiting doses of tamoxifen administered to *Smarcd3*-F1CreERT2;*Rosa*R26R embryos at E5.5 label scattered clusters of cells throughout the heart at E10.5. (**I**) Clonal analysis with the *Smarcd3*-F1CreERT2;*Rosa*Confetti line (E5.5 label, harvested at E10.5) shows that a single *Smarcd3*-F1+ progenitor can populate the left ventricle (YFP, arrow heads) or the left atrium (red fluorescent protein, asterisk). Scale bars: (**A** and **D**), 500 μm (**B** and **E**), 200 μm (**I**), 50 μm. en, endocardium; LV, left ventricle; LA, left atrium; m, myocardium; OFT, out flow tract; p, pericardium; RA, right atrium; RV, right ventricle.

The following figure supplements are available for figure 5:

**Figure supplement 1**. Generation and characterization of *Smarcd3*-F1CreERT2 and *Smarcd3*-F6CreERT2 mice.

**Figure supplement 2**. Additional characterization of *Smarcd3*-F1CreERT2 mice.

**Figure supplement 3**. Additional characterization of *Smarcd3*-F6CreERT2 mice.

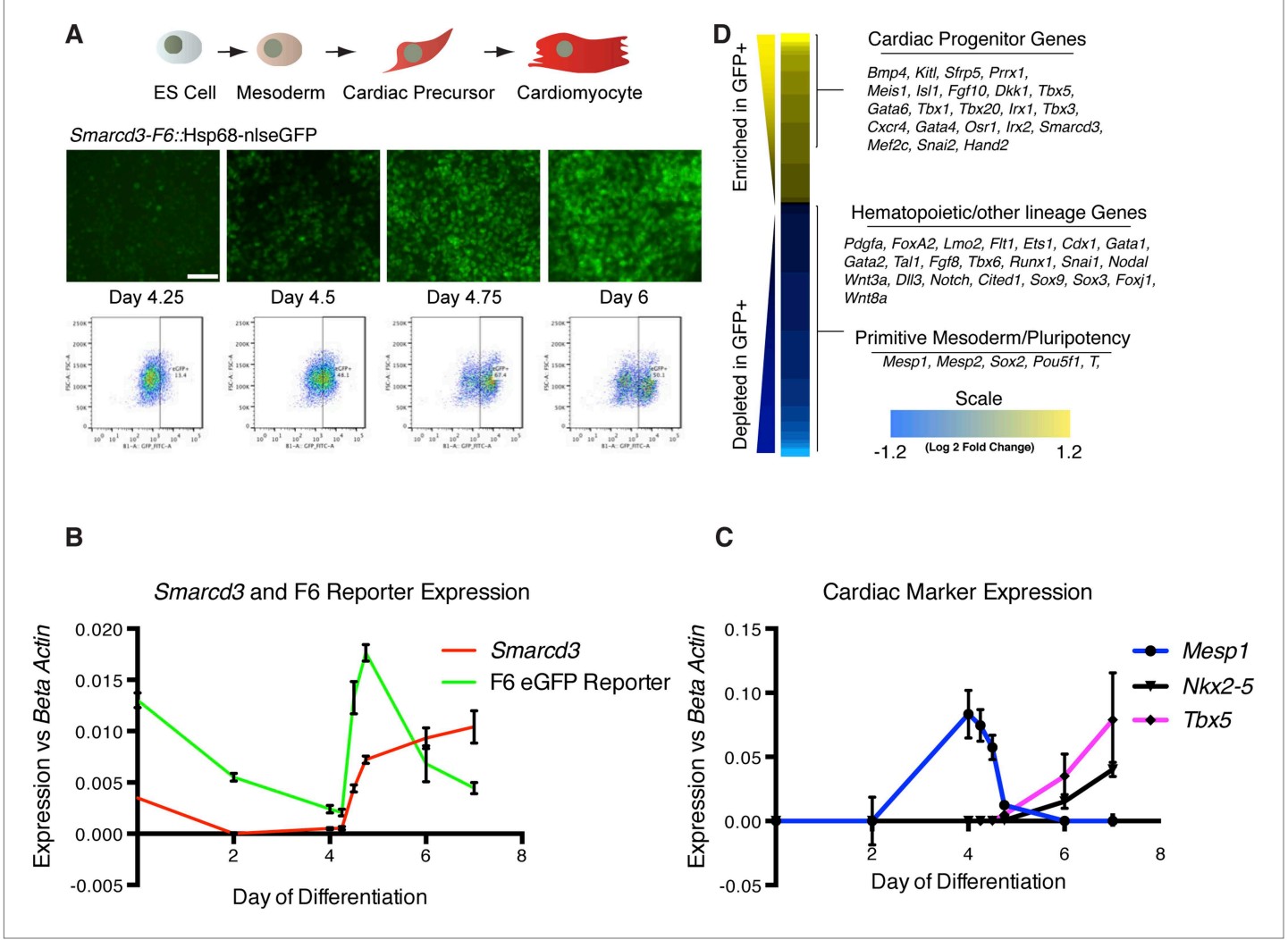

**Figure 6**. *Smarcd3*-F6 enhancer labels an early cardiac progenitor in differentiating ES cells. (**A**) Schematic of cardiac differentiation protocol and representative images at indicated time points during differentiation of *Smarcd3*-F6nlsEGFP mESCs. (**B**) Time course of gene expression during cardiac differentiation. Low expression of *Smarcd3* as well as *GFP* is detectable in ES cells. Expression decreases over the course of the differential protocol but is rapidly induced shortly after day 4. (**C**) Expression of *Mesp1* peaks between day 3-day 4 of the differentiation protocol. Expression of *Nkx2-5* and *Tbx5* begins later at day 5. Values shown are the mean plus SEM for 4 independent experiments, each performed in triplicate (**D**) Heatmap showing differential gene expression in GFP + compared to GFP- sorted cells. Many genes involved in cardiac progenitor development are enriched in the GFP + population. Markers of primitive mesoderm and of hematopoietic and other cell lineage development are enriched in the GFP- population. Values are log$_2$ fold change and are clipped at 1.2. Analysis is based on three biological replicates. Yellow = higher in GFP positive population, blue = higher in GFP negative population.

The following source data and figure supplement are available for figure 6:

**Source data 1**. Complete list of differentially expressed genes meeting strict FDR of 0.02.

**Figure supplement 1**. Generation and characterization of *Smarcd3*-F6nlsEGFP mESC line.

(Day 3-Day 4) while expression of the cardiac precursor markers *Nkx2-5* and *Tbx5* initiate about a day later (*Figure 6C*). Expression of *Smarcd3* as well as the *Smarcd3*-F6 reporter are detectable in ES cells and gradually decline until both are sharply induced at Day 4. At the same time, *Mesp1* expression is rapidly downregulated (*Figure 6B*). As a result, *Mesp1* and *Smarcd3* mRNA are largely non-overlapping, temporally. We isolated total RNA from three biological replicates of sorted cells shortly after the initiation of reporter activity at the mesoderm stage (D4.75) and compared the gene expression profiles

of GFP + to GFP- cells by RNA-seq (*Figure 6D*). GFP + cells expressed genes associated with early cardiovascular progenitors (e.g. *Hand2, Gata4, and Meis1*), while GFP- cells expressed genes associated with hematopoietic and other lineages. Among the GFP + population, we attempted to identify markers that were unique to either cardiomyocyte or endocardial progenitors. While cell type-specific markers for differentiated cells are well documented (eg. contractile proteins for cardiomyocytes vs *Nfatc1* for endocardial cells) such markers are less well defined at the early stages when we are isolating mRNA. In addition, our differentiation protocol pushes cells towards a cardiomyocyte fate, thus we may not expect to see any endocardial cell markers. One gene of interest that appears to be depleted in our GFP + population/enriched in our GFP- population is *Tal1/Scl*. This gene had been implicated as a critical component of endocardial morphogenesis (*Bussmann et al., 2007*) and more recently has been shown to repress cardiomyogenesis in yolk sac vasculature and endocardium (*Van Handel et al., 2012*). A complete list of differentially expressed genes meeting a strict false discovery rate (FDR) of 0.02 (2%) is presented in *Figure 6—Source data 1*. Overall, these results indicate that cells labeled by the *Smarcd3*-F6 enhancer, both in vivo and in vitro, represent an early cardiac progenitor population.

One class of clones identified in our *Mesp1Cre*-MADM clonal analysis included twin spots within the interventricular septum (IVS) between the left and right ventricles (*Figure 7A–D* and *Figure 2*). These twin spots formed a sharp boundary within the IVS, reminiscent of compartment boundaries classically defined in the *Drosophila* wing imaginal disc and mammalian midbrain-hindbrain boundary (*Lawrence and Struhl, 1996*; *Dahmann and Basler, 1999*). To determine if and when this boundary between the right ventricle and left ventricle is established, we performed temporally regulated genetic fate-mapping to mark early cardiac progenitors and followed their contribution later in the mature IVS. In order to mark cells that would contribute to the right ventricle, we used the *Mef2c-AHF* enhancer (*Verzi et al., 2005*) to drive expression of a fusion of the Dre recombinase (*Anastassiadis et al., 2009*) to the tamoxifen inducible ERT2 protein (*Mef2cAHFDreERT2*). We found that this enhancer element is expressed very early in the embryo, in a domain that appeared to partially overlap with that of *Smarcd3* (*Figure 3C,H*). Previous work using a constitutive Cre recombinase expressed under the control of the *Mef2cAHF* enhancer suggested that endothelial and myocardial components of the outflow tract, right ventricle, and IVS are derived from this population (*Verzi et al., 2005*). To label the complementary population of progenitors that contribute predominantly to the left ventricle, we targeted a tamoxifen inducible Cre recombinase to the *Tbx5* locus (*Tbx5CreERT2*) (*Figure 7—figure supplement 1A–B*). In the looped heart, expression of *Tbx5* is restricted to the left ventricle and atria with very little expression in the right ventricle or outflow tract (*Bruneau et al., 1999*). Early labeling of *Tbx5+* cells marked cells that were predominantly restricted to the left ventricle and atria (*Figure 7E–F and 7I–L*); scattered surface labeling on the right ventricle (*Figure 7—figure supplement 1I–J*) was also seen with early tamoxifen injection, but myocardial labeling was limited to the LV up to the junction with the RV at the IVS. Earlier observation at E8.5 following a pulse of tamoxifen at E6.5 revealed labeling in a restricted population of cells within the presumptive left ventricle (*Figure 7—figure supplement 2*), suggesting that the Tbx5 lineage restriction arises early and is not a consequence of later sorting out of cells. Labeling *Mef2cAHF* + cells at E6.5 marked a complementary population of cells that were largely restricted to the right ventricle and outflow tract (*Figure 7G–H* and *Figure 7—figure supplement 3A*). Using an intersectional reporter that responds to the combined activity of Cre and Dre we confirmed that these early *Mef2cAHF* + cells are also within the *Smarcd3*-F1 lineage (*Figure 7M* and *Figure 7—figure supplement 3B–C*). The expression of *Tbx5* and the *Mef2AHF* enhancer appears to label complementary populations that are established prior to morphogenesis, which correspond, in part, to left and right ventricular precursors, respectively.

Given the seemingly complementary lineage contributions of early *Tbx5* and *Mef2cAHF* expressing progenitors to the mature heart, we sought to map their contributions and overlap within the interventricular septum. We performed simultaneously lineage tracing of *Tbx5+* and *Mef2cAHF* + cells using separate reporters for Cre (*RosamTmG*) and Dre (*RosanKmB*) activity and found that the two populations of labeled cells appeared largely mutually exclusive, except perhaps a narrow stripe of cells at the border between the forming left and right ventricles (*Figure 7N*). We confirmed the existence of a small population of double positive cells early in development using the intersectional Cre/Dre reporter described previously. Labeling at E6.5 marked a small population of cells that would eventually go on to form a narrow ring of cells at the border between the left and right ventricles in the forming IVS (*Figure 7O–P* and *Figure 7—figure supplement 3D*). In summary, genetic lineage labeling of the

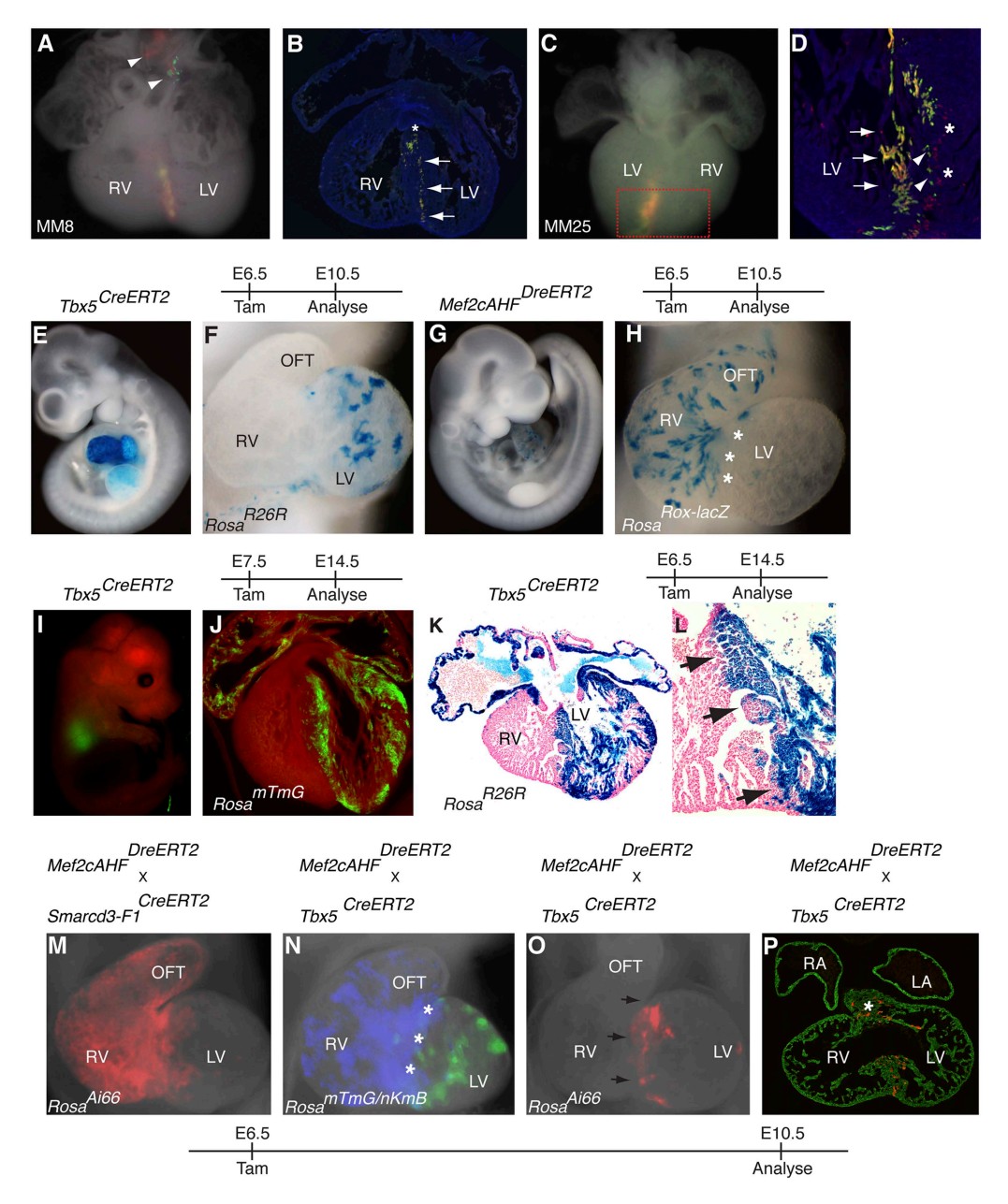

**Figure 7**. Early establishment of a boundary between the right and left ventricle at the interventricular septum. (**A**) Ventral view of yellow septal clone (embryo ID MM8) with additional red and green twin-spots in outflow tract (arrow heads). (**B**) Section through heart reveals sharp septal boundary of clone (arrows) with an extension of cells at top of septum into RV (asterisk). (**C**) Ventral view of large septal clone (embryo ID MM25) originating from left ventricle. (**D**) Section through red-boxed area reveals large yellow clone (arrows) extending from apex of left ventricle towards the top of the interventricular septum. An additional clone of red cells (asterisks) is directly adjacent to the yellow clone. The green twin spot is located just medial to the red twin spot (arrowheads). All clones appear to be originating from the apex of the left ventricle. (**E–F**) Tbx5+ cells labeled at E6.5 and observed at E10.5 contribute to left ventricle and atria. (**G–H**) Mef2cAHF + cells labeled at E6.5 and observed at E10.5 contribute to specific anterior heart field structures, including the right ventricle and outflow tract. Note sharp boundary at future site of interventricular septum (IVS, asterisks). (**I–J**) Tbx5+ cells labeled at E7.5 and observed at E14.5 for mTmG. (**K–L**) Tbx5+ cells labeled at E6.5 and observed at E14.5 for R26R. A sharp boundary at IVS between the left and right ventricles is present following early labeling of Tbx5+ cells. (**M**) Smarcd3-F1/Mef2cAHF double positive cells were labeled at E6.5 and observed at E10.5 using the intersectional reporter, Rosa^Ai66. Labeled

*Figure 7. Continued on next page*

*Figure 7. Continued*

cells contribute to right ventricle and outflow tract with a minor population of cells extending into the left ventricle. (**N**) Tbx5+ cells were labeled at E6.5 and their lineage followed using the *Rosa*$^{mTmG}$ Cre-reporter (green). Mef2cAHF + cells were also labeled at E6.5 and their lineage followed using the *Rosa*$^{nKmB}$ Dre-reporter (blue). Note largely non-overlapping Tbx5 and Mef2cAHF derived lineages in left and right ventricles, respectively, except perhaps a small area of overlap at the forming interventricular septum (asterisks). (**O**) Tbx5/Mef2cAHF double positive cells were labeled at E6.5 and observed at E10.5 using the intersectional reporter, *Rosa*$^{Ai66}$. A narrow ring of labeled cells is present between the left and right ventricles. (**P**) Sections confirm a restricted population of labeled cells within the interventricular septum and superior aspect of the ventricular chamber (asterisk). Red, TdTomato; Green, Tropomyosin.

The following figure supplements are available for figure 7:

**Figure supplement 1**. Generation of a multi-use Tbx5 allele.

**Figure supplement 2**. Early labeling of Tbx$^{5CreERT2}$ lineage.

**Figure supplement 3**. Additional characterization of early Mef2cAHF and Tbx5 lineages.

early*Tbx5*+ and *Mef2cAHF* + precursors along with our clonal analysis of early *Mesp1*+ progenitors that contribute to the IVS suggest that a compartment boundary between the future left and right ventricles is established early, prior to cardiac morphogenesis, and that early progenitors positive for both *Tbx5* and *Mef2cAHF* will go on to contribute specifically to the forming IVS.

## Discussion

The existence of a multipotent cardiac progenitor that can contribute to all anatomic and cellular components of the mature heart has been predicted, but the identity and origins of such a progenitor has remained undefined. Our findings, summarized in **Figure 8**, pinpoint the existence of specified cardiac precursors in gastrulating mesoderm, and also highlight an unanticipated early segregation of first and second heart field progenitors in their contribution to distinct chambers of the developing heart. Further, we define the orderly progression of gene expression that parallels the commitment of nascent mesoderm to a cardiovascular fate. In particular, the expression of *Smarcd3* labels a population primarily comprised of the earliest specified precursors, which can be identified in vivo and in ES cell-derived differentiating cells.

Our results show that mesoderm is rapidly specified at or before E6.0 to E7.5 into discrete fates that anticipate anatomical localization. Retrospective lineage tracing using a cardiac marker suggested the existence of a precursor that could contribute to all chambers of the heart (**Meilhac et al., 2004b**). Our results, and those of **Lescroart et al. (2014)**, suggest that such a precursor could only exist prior to gastrulation, likely in the epiblast or shortly thereafter, and that early mesoderm is rapidly patterned such that specific cells are shunted towards a unique path that assigns them to a specific cardiac compartment. Whether these mesodermal cells retain plasticity and could adopt different fates if transplanted elsewhere in the embryo is not clear. We also cannot at this point distinguish between a scenario where *Mesp1*+ cells are pre-patterned or the gradual restriction of *Mesp1*+ cells to different lineage fates during gastrulation in response to signaling cues. Understanding which molecular cues induce the emergence and patterning of this population of early cardiac precursors will be of considerable interest, particularly in the context of regenerative cell therapy, and especially given the apparent heterogeneity of early cardiac progenitors.

The *Smarcd3*-expressing cardiac precursor population is already specified into separate populations that will contribute to the left ventricle and atria, as marked by *Tbx5*, and the right ventricle and outflow tract, as labeled by the *Mef2cAHF* enhancer. These results are consistent with genetic tracing of cardiac precursors expressing *Hcn4*, which contribute to the left ventricle and conduction system (**Später et al., 2013**), and we additionally show, at both a population and single cell level, that both left and right ventricular progenitors are independently established early in development and are separated by a compartment boundary. This compartment boundary is derived from a unique population of early progenitors that express *Tbx5* as well as a subpopulation of cells co-expressing the *Mef2cAHF* enhancer, suggesting that a relatively small number of cells may be important in

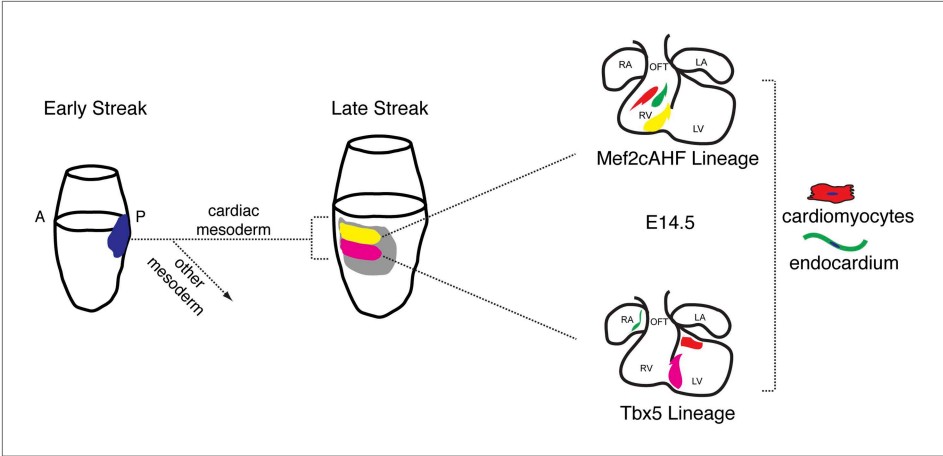

**Figure 8**. Summary of patterning and specification of early gastrulating mesoderm. Clonal analysis reveals early patterning of gastrulating mesoderm including the segregation of cardiac vs non-cardiac mesoderm. Among cardiac mesoderm, progenitors for the two heart fields diverge soon after the initiation of gastrulation and rapidly become specified into discrete populations of committed precursors. Expression of *Smarcd3* begins prior to that of other known markers of cardiac progenitors (*Nkx2-5, Tbx5, and Isl1*) and an enhancer of *Smarcd3* (*Smarcd3*-F6) marks the earliest cardiac-specific progenitor population. Expression of the *Mef2AHF* enhancer and *Tbx5* further subdivides this early population into first vs second heart field progenitors. Inducible genetic lineage tracing along with clonal analysis predicts the existence of a compartment boundary between the future left and right ventricles that is established prior to cardiac morphogenesis.

establishing and/or maintaining this border between the left and right ventricles. Understanding the molecular and cellular basis of this compartment boundary, and its importance in ventricular septation, will be of great relevance to congenital heart defects. This knowledge will also be relevant for disease modeling from stem cells, or for regenerative strategies, since knowing whether a progenitor cell is committed to become a right ventricular cardiomyocyte vs an atrial cardiomyocyte, for example, could be critical depending upon the anatomic or cellular defect one is attempting to correct.

In organisms with a simpler heart, such as zebrafish or *Ciona* (**Stolfi et al., 2010**; **Tolkin and Christiaen, 2012**; **Stainier et al., 1993**; **Keegan et al., 2004**), there is also a very early specification of cardiac precursors. Indeed, fate mapping in the chick embryo suggests distinct origins of the outflow tract vs the remainder of the heart, but does not indicate a compartmentalization of left and right ventricular progenitors, nor what molecular identity early cardiac precursors adopt (**Camp et al., 2012**). Our data suggest that the mouse embryo, despite the complexity of its developing heart, has a very early patterning and allocation of mesodermal cells to specific cardiac fates and anatomical derivatives. In addition, it has been hypothesized that simpler organisms have integrated cardiac and pharyngeal mesoderm into a single lineage, the so-called cardiopharyngeal mesoderm (**Stolfi et al., 2010**). Retrospective clonal analysis in the mouse (**Lescroart et al., 2010**) suggests a conservation of shared cardiac and pharyngeal lineages; caveats applicable to chamber-specific lineage allocation, including the timing of clone induction and the promoter used to visualize clones, remain. Lescroart and colleagues more recently reported rare instances of *Mesp1+* clones co-labeling head muscles and the heart (**Lescroart et al., 2014**). In our lineage labeling experiments, using the MADM system, we do not see convincing evidence for a common heart and head muscle progenitor. Additionally, *Smarcd3+* precursors contribute to heart but not facial muscle. While our collective data do not support a common precursor for heart and facial muscle, we cannot definitively rule out a rare population of common progenitors.

Having a well-defined understanding of the developmental origins of patterned organs is critical for the development of regenerative strategies, and for our understanding of the basis of congenital malformations. For the future development of cellular therapies, the knowledge that molecularly distinct populations of progenitors contribute to distinct anatomic regions of the heart will guide the selection of the most appropriate cellular and molecular signature. More broadly, our results suggest a rapid and precise patterning of progenitor populations during gastrulation.

## Materials and methods

### Mice

The *Mesp1^Cre* knock-in mice (*Saga et al., 1999*) were obtained from Yumiko Saga. *Mef2cAHF*-lacZ mice (*Dodou et al., 2004*) were obtained from Brian Black. *Isl1^nLacZ* knock-in mice (*Sun et al., 2007*) were obtained from Sylvia Evans. MADM11^TG/TG, MADM11^GT/GT mice (*Hippenmeyer et al., 2010*), ROSA^R26R mice (*Soriano, 1999*), ROSA^mTmG mice (*Muzumdar et al., 2007*), ROSA^Ai66 mice (Allen Brain Institute strain B6;129S-*Gt(ROSA)26Sortm66.1(CAG-tdTomato)Hze*/J; JAX stock number 021876), *Flk1^GFP* knock-in mice (*Ema et al., 2006*), and Sox2::Cre transgenic mice were obtained from Jackson Laboratory. ROSA^RoxlacZ mice (*Anastassiadis et al., 2009*) were generated from cryopreserved embryos purchased from MMRRC. Standard tamoxifen induction was done by injecting 3 mg/40 gram of body weight of tamoxifen dissolved in sesame oil intraperitoneally. Low dose tamoxifen induction was done by injecting 1/10th of this concentration (0.3 mg/40 gram body weight). Clonal analysis of *Mesp1* mesodermal progenitors was performed by crossing *Mesp1^Cre*;MADM11^TG/TG with MADM11^GT/GT mice. Lineage analysis of *Smarcd3* cardiac progenitors was performed by crossing *Smarcd3*-F1CreERT2 mice with ROSA^R26R, ROSA^mTmG, or ROSA^Confetti mice followed by administration of tamoxifen (3 mg/40 grams of pregnant dam's body weight (*Hayashi and McMahon, 2002*) ) at E5.5 or E6.5. Lineage analysis of *Mef2cAHF*-expressing progenitors was performed by crossing ROSA^Rox-lacZ and *Mef2cAHF^DreERT2* transgenic mice followed by administration of tamoxifen at E5.5 or E6.5. Lineage analysis of *Tbx5*-expressing progenitors was performed by crossing *Tbx5^CreERT2* knock-in mice and ROSA^R26R or ROSA^mTmG mice followed by administration of tamoxifen at indicated times.

### Cloning and generation of transgenic mice

The *Smarcd3*-F1 fragment spanned 8796bp upstream of the start codon of *Smarcd3* (Chr5:24,107,677-24,116,473) and was cloned using bacterial recombineering from BAC bMQ133n21 into pENTR1A (Invitrogen, Carlsbad, CA). The pWHERE plasmid (InvivoGen) was digested using a blunt cutter (SmaI) and an RFA 'C' Gateway cassette (Invitrogen) was inserted 5′ to the promoterless nls-LacZ reporter gene to make a destination vector (pWHERE-DV). For construction of pWHERE-DV-CreERT2, a Gateway RFA 'B' cassette was amplified by PCR and cloned into the AvrII site of pWHERE using Cold Fusion cloning (System Biosciences Inc.). The resulting plasmid was digested with XhoI and NheI to remove the nls-LacZ reporter and a cassette encoding CreERT2 (*Feil et al., 1997*) was amplified and inserted using Cold Fusion Cloning. Construction of the *Mef2cAHF^DreERT2* allele will be described in detail elsewhere. Briefly, the CAGEN-DV plasmid (Devine and Bruneau, unpublished) was digested with SpeI and XbaI to remove the CAGGS (Chicken Beta-Actin promoter with CMV enhancer) promoter and the resulting ends blunted. The Mef2c-F6/frag3 plasmid (*Dodou et al., 2004*) was digested with XhoI and SalI to isolate the 3970 bp cardiac enhancer fragment. The resulting ends were blunted and the two blunt-ended fragments ligated together to make the destination vector (Mef2c-F6-DV). A Gateway compatible entry clone for DreERT2 was constructed by PCR stitching of Dre (minus the nls) from plasmid DNA containing a codon-improved version of Dre (a generous gift of Francis Stewart, Biotechnology Center TU, Dresden, Germany) and ERT2 (*Feil et al., 1997*) separated by a short linker se quence (*Hunter et al., 2005*) to generate DreERT2. Upon LR recombination with entry clones and destination reporter plasmids, the final vectors were restriction mapped and verified by DNA sequencing. Multiple founders were examined for each transgene. All experiments using mice were reviewed and approved by the UCSF Institutional Animal Care and Use Committee and complied with all institutional and federal guidelines.

For construction of the *Rosa^nKmB* fluorescent Dre-reporter allele, a nuclear localized mKateV5 (nlsmKateV5) followed by a rabbit globin polyA sequence was cloned between ROX recombination sites downstream of a CAGGS promoter. A Gateway RFA destination cassette (including a rabbit globin polyA sequence) was subsequently cloned downstream to make pCAGGS-nK-DV. A SacI-SalI fragment containing an FRT-pGK-FRT cassette from pK11 (plasmid courtesy of Gail Martin) was cloned downstream of the Gateway RFA-rabbit globin polyA sequence to make pCAGGS-nK-DV-FRT-pGK-FRT. A Gateway entry clone containing a membrane localized TagBFP-FLAG was inserted into pCAGGS-nK-DV-FRT-pGK-FRT using LR recombination. The entire construct was excised using AscI and PacI and cloned into Rosa26PAm1 (Addgene plasmid# 15,036) for targeting to the *Rosa26* locus. The construct was linearized with SgfI and electroporated into

E14 mouse ES cells. Drug resistant clones were screened by long range PCR using the following primers:

| | |
|---|---|
| 5' homology arm: | ROSA1 5'-CCACTGACCGCACGGGGATTC-3' (in genomic) |
| | ROSA7 5'-GGGGAACTTCCTGACTAGGG-3' (in FRT) |
| 3' homology arm: | ROSA2 5'-TCAATGGGCGGGGGTCGTT-3' (in CAGGS) |
| | ROSA5 5'-GGGGAAAATTTTTAATATAAC-3' (in genomic) |

Positive clones were subsequently screened by qPCR for single copy insertions of the pGK-Neo cassette. Following verification of correct targeting and karyotyping, two positive ES cell clones were expanded and injected into blastocysts for generation of mice. Chimeric founders were crossed to C57B6 lines to confirm germline transmission. The pGK-Neo cassette was eventually removed by breeding mice to Act^Flpe expressing mice.

## Cloning and generation of TARGATT transgenic knock-in mice

The *Smarcd3*-F6 fragment, including approximately 2.5 kb of the 5' end of the *Smarcd3*-F1 promoter fragment was cloned by PCR into pENTR1A. Modified pWHERE-DV and pWHERE-DV-CreERT2 plasmids were generated by inserting an Hsp68 minimal promoter into the XhoI restriction site using Cold Fusion Cloning. Upon LR recombination with entry clones and destination reporter plasmids, the final vectors were restriction mapped and verified by DNA sequencing. For generating TARGATT constructs, the PacI fragment from the final construct was subcloned into a PacI digested pBT346.3 plasmid (Applied Stem Cells). DNA was purified and injected along with mRNA for the *Phi31o* transposase according to manufacturer's protocol.

## Cloning and generation of *Tbx5* knock-in mice

A 129 BAC clone (67H11, RPCI-22 mouse BAC library) containing the entire mouse *Tbx5* locus was obtained from GeneService (UK). Briefly, the cDNA for CreERT2 (*Feil et al., 1997*), followed by an IRES element (BamHI to NcoI, from pIRES2-EGFP), a Kozak consensus sequence and a 2X FLAG epitope sequence in frame with the translational start of endogenous *Tbx5* was inserted between the FspI and NcoI sites of exon 2 of *Tbx5*, upstream of the endogenous translation start site. At 40bp downstream of exon 2, cloning sites (NotI and AflII) were added and a PGK-EM7-Neo-polyA cassette, flanked by Frt sites (from PL451), was inserted for positive selection. The entire cassette, as well as 5 kb upstream of exon 2 (ClaI to FspI), and 6 kb downstream of exon 2 (40 bp downstream of exon 2 to BclI) were cloned into a modified pBS containing a 5' DTA negative selection cassette (from pRosa-26-1). The targeting vector was linearized by SalI digestion and electroporated into embryonic stem cells, and G418-resistant clones were tested for correct gene targeting by Southern analysis using 5' and 3' (not shown) probes external to the targeting vector.

The following primers were used for the 5' probe:
probeA-F1: 5'-GGCCACTGATGGTGTAGAAGCAAC-3'.
probeA-R1: 5'- GTAGAGAGAAAGGCCATTCGGTCTG -3'.
The following primers were used for the 3' probe:
probeB-F1: 5'-GGGCCATTAGATCACCCTCATTCTG-3'.
probeB-R1: 5'- AACTCTGTGTATAAGGGCACTTCCC -3'.

Following verification of correct targeting and karyotyping, positive ES cells were expanded and injected into blastocysts for generation of mice. Chimeric founders were crossed to C57B6 lines to confirm germline transmission. The following primers were used for initial genotyping:

| | |
|---|---|
| 5' end of targeted locus: | Tbx5CreF1: TATGTCGCTAGACACTCTCC |
| | Tbx5CreR1:CCGGCAAACGGACAGAAGCA |
| | Knock-in = 226 bp, WT = no band |
| 3' end of the targeted locus: | Tbx5CreNeoFor1: ACTGTGCCTTCTAGTTGCCAGC. |
| | Tbx5CreWT: Rev: AAAGTGGATTGGGATAGAGTGG |
| | Knock-in = 470 bp, WT = no band |

*Table. Continued on next page*

*Table. Continued*

| After mating to B-actin-FlpE mice to remove the Neo cassette: | Tbx5NeoFlp'DF1: ACAACCATGGACTACAAGGACG |
| --- | --- |
| | Tbx5CreWT Rev: AAAGTGGATTGGGATAGAGTGG |
| | Knock-in = 420 bp, WT = no band |
| Mice were maintained on a C57B6 background after crossing to various reporter mice. Inheritance of the allele was confirmed by PCR | Tbx5Exon2WTFor2: ATACAGATGAGGGCTTTGGCCTGG |
| | Tbx5CreWTRev: AAAGTGGATTGGGATAGAGTGG |
| | WT band = 290 bp, CreRT2 band = 360 bp |

## Generation and culturing of *Smarcd3*-F6nlsEGFP mESC line

The *Smarca4*[FLAG] knock-in ES cell line (*Attanasio et al., 2014*) was used for targeting of the *Smarcd3*-F6-*Hsp68-nlsEGFP* construct to the *Hipp11* locus. Briefly, a modified shuttle vector containing a polylinker including PacI, XhoI, SacII, and flanking AscI sites was purchased from IDT. A pGKNeo selection cassette was subcloned from the pL451 plasmid using XhoI and SacII into the modified shuttle vector. A PacI fragment including flanking H19 insulator sequences, the *Smarcd3*-F6 enhancer, an Hsp68 minimal promoter, *nlsEGFP* coding sequence, WPRE mRNA stablilization sequence, and EF1alpha poly A sequence was subcloned into the modified shuttle vector. The entire reporter-selection construct was cloned into the *Hipp11* targeting vector (*Hippenmeyer et al., 2010*) using AscI. The targeting vector was linearized using ApaI and electroporated into ES cells. Following G418 selection, correctly targeted clones were screened by PCR and Southern blotting. For culturing, ES cells were maintained in 2i + LIF media.

## Fluorescent activated cell sorting and RNA sequencing

Directed cardiomyocyte differentiations were performed as previously described (*Wamstad et al., 2012*) using the *Smarcd3*-F6nlsEGFP mESC line with minor modifications. 18 hours after plating and cardiac induction (with VEGF, Fgf10, and Fgf2), supernatant was collected and 0.22 μm filtered. Cells were then washed with PBS (w/o $Ca^{2+}/Mg^{2+}$), dissociated from plates using TrypLE (Gibco), resuspended in filtered supernatant, and placed on ice. $GFP^+$ and $GFP^-$ populations were subsequently sorted into RNAprotect cell reagent (Qiagen) using a BD FACSAria II flow cytometer. RNA was then purified from each population using the RNeasy Mini kit (Qiagen). Stranded RNA-seq libraries were then prepared using the Ovation Mouse FFPE RNA-Seq Multiplex System (NuGEN) and sequenced on an Illumina HiSeq 2000. Three biological replicates for each population (GFP+ and GFP-) were obtained and analyzed by RNA-sequencing.

## Bioinformatics analysis

Sequence reads were aligned to the mm9 (mouse) assembly with Tophat 2 (*Kim and Salzberg, 2011*), using Ensembl version 67 exon annotation. Differential expression and variance-corrected log fold change was calculated using the program 'DefinedRegionDifferentialSeq' in USeq version 8.6.4 (http://useq.sourceforge.net/). In order to report gene-level counts, the highest-total-read-count transcript was reported for each gene, resulting in gene level annotation only. The final heatmap reports all genes/nonCode elements where the Benjamini-Hochberg FDR-corrected value (*Benjamini and Hochberg, 1995*; *Nix et al., 2008*) (false discovery rate) was less than 0.02 (i.e. 2% false discovery) in the comparison of (GFP+) vs (GFP-).

## In situ *hybridization*, immunostaining and LacZ staining

Embryos were processed as previously described (*Wythe et al., 2013*). For lacZ embryos, beta galactosidase activity was detected using Salmon Gal (Sigma) prior to processing for in situ hybridization. Immunostaining of MADM samples for GFP and Myc was as previously described (*Zong et al., 2005*). Antibodies used on cryosections include: Rabbit anti-GFP (Invitrogen, 1:1000), Goat anti-Myc (Novus, 1:200), Rat anti-CD31 (BD Pharmingen, 1:100), Mouse anti-tropomyosin (DSHB clone CH-1, 1:50). Whole-mount lacZ and indirect immunofluorescent images were obtained using a Leica dissecting microscope and camera with the Leica LAS Montage extended focus function. Confocal images were obtained on a Nikon ECLIPSE Ti 2000 confocal microscope with a Yokogawa CSU-X1 spinning disk and Hamamatsu ImagEM CCD camera. Images were processed using Volocity software (Perkin Elmer). All images, including immunofluorescent, in situ hybridization, and LacZ images are representative images. At least 5 embryos (in situ hybridization and LacZ) or 3 independent sections (immunofluorescence)

were examined for each experiment. Images shown represent average or representative expression levels.

## Mouse experiments

All mouse protocols were approved by the Institutional Animal Care and Use Committee at UCSF.

## Statistical analyses

In the *Mesp1*[Cre]-MADM mice, heart labeling results from a Cre-mediated chromosomal translocation that occurs within a narrow developmental time window of Cre expression, between the initiation of gastrulation and shortly there after (E6.0 and E7.0). Importantly, this translocation event does not happen in the absence of Cre-recombinase (our observations and previously published (*Tasic et al., 2012*)) and there is no visible labeling prior to translocation. We have empirically defined the frequency of this event by measuring two different variables: (1) the total number of cells that can undergo this translocation and (2) the observed frequency of labeled clones. The total number of cells that could undergo translocation was determined by counting the number of cells that had recombined a Cre-dependent reporter (*Rosa*[td-Tomato]) at the end of gastrulation (E7.5). FACS analysis determined that this number was ~850 cells (~1/3 of the total number of cells in the embryo at this time (*Figure 1—figure supplement 1C*)).

Given the fact that we observed a total of 96 clones across 38 embryos, our clonal sampling represents 11% (or 96 clones / 850 Mesp1+ cells) of the total *Mesp1*+ population (assuming a random sampling). Thus a rare subpopulation, such as a common progenitor for both the right and left ventricles, would be missed only if it represented less than 10% of the total Mesp1 population.

We also measured the observed frequency of labeled clones at two different time points (E8.5 and E14.5) to determine if the frequency of labeled clones changed over time (either increased or decreased) as might happen with additional recombination outside of our time window (E6.0–E7.0) or with selective loss of a twin spot via apoptosis (*Figure 1—figure supplement 1D–E*). Although the number of samples we have analyzed at E8.5 is small, we detected no significant change in the observed frequency of labeled clones at the two time points analyzed and thus conclude that recombination frequency is stable over developmental time and there is no gain or loss of clones outside our narrow labeling time window.

Based on the number of observations made (n = 96) and the fact that we did not observe a common progenitor (number of successes = 0), we calculated the upper and lower bounds of a 95% and 99% confidence interval (CI) that a common progenitor does not exist using a binomial probability (Jeffreys interval) appropriate for instances when the number of successes is either very close to 0 or very close to 1.

$$from \begin{cases} I_{\frac{1-c}{2}}^{-1}\left(x+\dfrac{1}{2}, n-x+\dfrac{1}{2}\right) \\ 0 \\ 1 \end{cases} \quad 0 < \dfrac{1-c}{2} < 1$$

$$\dfrac{1-c}{2} \le 0 \; for -2 \le c-1 \le 0$$

$$to \begin{cases} I_{\frac{c-1}{2}+1}^{-1}\left(x+\dfrac{1}{2}, n-x+\dfrac{1}{2}\right) \\ 0 \\ 1 \end{cases} \quad 0 < \dfrac{c-1}{2}+1 < 1$$

$$\dfrac{c-1}{2}+1 \le 0 \; for -2 \le c-1 \le 0$$

x = number of successes, n = sample size, c = confidence interval $I_x^{-1}(a, b)$ is the inverse regularized beta function.

We used the R Package 'binom', version 1.1-1 for calculating these values. The calculated values for a 95% CI were: 0 (lower) and 0.01975768 (upper) and the calculated values for a 99% CI were: 0 (lower) and 0.03387941 (upper). Thus, given the number of observations we made, we can be quite confident that a common progenitor does not exist.

An additional level of confidence regarding the lineage relationship of clones in different anatomical regions of the heart can be gained by comparing the color combinations observed. For example, clone MM2 could be interpreted as one recombination event that gave rise to two clusters: one in the right ventricle and one in the left ventricle. However the color combinations seen (red/green twin spots in right ventricle and yellow/blank twin spots in left ventricle) exclude the possibility that these two clusters are derived from the same event. Likewise, for clone MM26, only two labeled patches are seen in the entire embryo (red twin spot in left ventricle and green twin spot in right atrium). Assuming there is no selective loss of twin spots (as we have determined above by measuring the clonal frequency at two different time points) we can conclude that these two clusters are derived from a single event.

## Acknowledgements

We thank Dario Miguel-Perez and Claire Cutting for assistance with mouse husbandry and dissections, Kiyonori Togi for expanding and breeding chimeric mice, Robert Chadwick (Gladstone Genomics Core) for RNAseq library preparation, Alexander Williams (Gladstone Bioinformatics Core) for RNAseq data analysis, Sylvia Evans for transgenic mice, Brian Black for transgenic mice, constructs and helpful discussions, and Alisha Holloway and Katherine Pollard for statistical discussions. B.G.B thanks William H. Younger, Jr. for continued support.

## Additional information

### Funding

| Funder | Grant reference number | Author |
| --- | --- | --- |
| National Heart, Lung, and Blood Institute | R01HL114948 | Benoit G Bruneau |
| National Heart, Lung, and Blood Institute | Bench to Bassinet Program U01HL098179 | Benoit G Bruneau |
| American Heart Association | Established Investigator Award | Benoit G Bruneau |
| California Institute for Regenerative Medicine | Clinical Scholar TG2-01160 | W Patrick Devine |
| American Heart Association | Scientist Development Grant 12SDG12060353 | Joshua D Wythe |
| National Center for Research Resources | C06 RR018928, to the J. David Gladstone Institutes | Benoit G Bruneau |
| National Institutes of Health | Predoctoral Training in Developmental Biology, T32 HD 007470 | Matthew George |
| National Institutes of Health | Training Grant 5T32-HL007731-20 | W Patrick Devine, Joshua D Wythe |
| Fumi Yamamura Memorial Foundation for Female Natural Scientists | Grants-in-Aid for Scientific Research (C) | Kazuko Koshiba-Takeuchi |
| Ministry of Education, Culture, Sports, Science, and Technology | Program for Young Independent Researchers | Kazuko Koshiba-Takeuchi |

The funders had no role in study design, data collection and interpretation, or the decision to submit the work for publication.

## Author contributions

WPD, Developed and analyzed the Mef2cAHFDreERT2 allele, Conception and design, Acquisition of data, Analysis and interpretation of data, Drafting or revising the article; JDW, Developed and analyzed the Mef2cAHFDreERT2 allele, Analyzed the Tbx5 lineage, Conception and design, Analysis and interpretation of data, Drafting or revising the article; MG, Conception and design, Acquisition of data, Analysis and interpretation of data; KK-T, Made the Tbx5CreERT2 mouse line, Conception and design, Drafting or revising the article; BGB, Conception and design, Analysis and interpretation of data, Drafting or revising the article

## Ethics

Animal experimentation: This study was performed in strict accordance with the recommendations in the Guide for the Care and Use of Laboratory Animals of the National Institutes of Health. All of the animals were handled according to approved institutional animal care and use committee (IACUC) protocols (#AN089375) of the University of California, San Francisco.

# Additional files

## Major dataset

The following dataset was generated:

| Author(s) | Year | Dataset title | Dataset ID and/or URL | Database, license, and accessibility information |
|---|---|---|---|---|
| Bruneau BG, Devine WP, George MR, Wythe JD | 2014 | Differential expression analysis for RNA-seq data between cardiac progenitor and uncommitted cells at day 4.75 | http://www.ncbi.nlm.nih.gov/geo/query/acc.cgi?acc=GSE58363 | Publicly available at NCBI Gene Expression Omnibus |

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
