## [Decision Letter]

Thank you for sending your work entitled “Early Patterning and Specification of Cardiac Progenitors in Gastrulating Mesoderm” for consideration at *eLife.* Your article has been favorably evaluated by Fiona Watt (Senior editor), Marianne Bronner (Reviewing editor), and 2 reviewers.

The Reviewing editor and the other reviewers discussed their comments before we reached this decision, and the Reviewing editor has assembled the following comments to help you prepare a revised submission.

With the goal of identifying developmental fields or boundaries in the heart, Devine and colleagues perform lineage-tracing studies on cardiac progenitor cells, defined by expression of mesodermal markers, in late pre-gastrulation/early gastrulation mouse embryos. Utilizing the mesodermal germ layer marking Mesp1 in conjunction with transcriptional elements from Smarcd3, which is expressed within the Mesp1 cell population, the authors make a compelling case that specification of cardiac progenitors occurs early, that this specification is concurrent for each heart field, and that there is not a common progenitor for primary and secondary heart fields. This is an important topic, as these stages are difficult to access technically and there are outstanding questions about the heterogeneity of cardiac progenitors. The number and variety of mouse strains employed for lineage tracing, some of which are generated for this manuscript, and the number of experiments performed, are impressive. The results suggest that the mammalian heart is compartmentalized at early life stages. Overall, this is an interesting and well executed study; however, a few of the conclusions need further support or clarification.

Major concerns:

1) The authors state that the Mesp-Cre is active in mesoderm from E6.0 to E7.0, but it is unclear how this is determined? The reference Saga 1999 does not seem to show this, as it is likely Cre Protein persists longer than the Mesp mRNA and the authors should mention this. Without a conditional Cre, the authors should explain more clearly how they know that new clones are not induced throughout development. Are E14.5 embryos optically clear enough (as compared to E8.5 embryos) to count total fluorescent clones and be certain that new cell labeling events do not continue? More explanation in the main text is requested.

2) The authors make a novelty claim that they have shown that individual cardiac progenitor cells give rise to multiple cardiac cell types. This is not shown conclusively, as they do not use adequate costaining to identify cells in their Figure 1 tissue sections. They should use antibodies against a myocyte marker and an endothelial marker in a 4-color stain of clonal progeny to show this.

3) In the text and staining in Figure 3. What is the evidence that Mesp1 and Smarcd3 mRNAs do not overlap? When possible, which it should be for reporters like GFP and lacZ, the authors should use fluorescence detection to colocalize markers in tissue sections. Then, arrows could be used to pinpoint cells in sections with different signatures to better make the authors' points.

4) The authors use a confetti approach with a Smarcd3 enhancer driving an inducible Cre. This is a useful way to confirm the MADM results, although there may be pitfalls with tamoxifen persistence, etc. The authors mention the results briefly that labeled cells within clones are contained in single chambers, but it is better to show the full dataset here to have an idea of how they made their conclusion.

5) The authors mention that early Tbx5+ cell labeling marks progeny corresponding to epicardial progenitors. However, they do not show cell type-specific markers to validate this colocalization. They need to do the costains or remove the statement.

[Editors' note: further revisions were requested prior to acceptance, as described below.]

Thank you for resubmitting your work entitled “Early Patterning and Specification of Cardiac Progenitors in Gastrulating Mesoderm” for further consideration at *eLife.* Your revised article has been favorably evaluated by Fiona Watt (Senior editor) and a member of the Board of Reviewing Editors. The manuscript has been improved but there are some remaining relatively minor issues that need to be addressed before acceptance, as outlined below:

Original Comment #4: The authors use a confetti approach with a Smarcd3 enhancer driving an inducible Cre. This is a useful way to confirm the MADM results, although there may be pitfalls with tamoxifen persistence, etc. The authors mention the results briefly that labeled cells within clones are contained in single chambers, but it is better to show the full dataset here to have an idea of how they made their conclusion.

Also, authors should indicate this small number of confetti animals examined in the text of their manuscript, and any limitation of their analysis, for the benefit of anyone who will attempt to repeat this work.

Original Comment #5: The authors mention that early Tbx5+ cell labeling marks progeny corresponding to epicardial progenitors. However, they do not show cell type-specific markers to validate this colocalization. They need to do the costains or remove the statement.

Further comment in response to authors' rebuttal: Disagree. Based on the image provided, the authors should remove the statement on contribution to epicardial progenitors. The red Tbx18 signal is equivalent to background – the signal and claim of costain is unlikely to convince the reader.

---

## [Author Response]

*1) The authors state that the Mesp-Cre is active in mesoderm from E6.0 to E7.0, but it is unclear how this is determined? The reference Saga 1999 does not seem to show this, as it is likely Cre Protein persists longer than the Mesp mRNA and the authors should mention this. Without a conditional Cre, the authors should explain more clearly how they know that new clones are not induced throughout development. Are E14.5 embryos optically clear enough (as compared to E8.5 embryos) to count total fluorescent clones and be certain that new cell labeling events do not continue? More explanation in the main text is requested*.

The reviewers raise an important point with respect to the timing of clone induction. While we have not used a conditional Cre allele to control the timing of Cre activity, we believe there are several lines of evidence that support the conclusion that *Mesp1*^*Cre*^ is active (and thus clones are induced) over a very narrow temporal window (from ∼E6.0-E7.5). First, we have performed *in situ* hybridization for *Cre* mRNA in *Mesp1*^*Cre*^ embryos at multiple stages and find that by the late head fold stage (LHF) expression of Cre mRNA is largely restricted to the allantoic base and allantoic membrane with minimal expression in area of the forming cardiogenic mesoderm (Figure 1—figure supplement 1), supporting the idea that *Cre* mRNA and thus Cre activity is extinguished after E7.5.

While staining for Cre protein would be the preferred method for determining protein perdurance, we have been unable to identify a suitable antibody for Cre immunostaining despite attempts with several different antibodies on both whole-mount embryos and tissue sections. Second, in the original reference describing the expression and requirement of *Mesp1* in early gastrulating mesoderm (38) they look both at *Mesp1* mRNA (their Figure 3) as well as *beta*-galactosidase activity in a knock-in allele (their Figure 3). Even with the perdurance of *beta*-galactosidase enzyme activity, the authors see minimal X-gal staining after E7.75. If we assume that *beta*-galactosidase activity likely persists as long or longer than Cre protein activity, it is unlikely that clones are induced after ∼E7.5. Third, we have counted the number of clones in embryos at E8.5 and at E14.5 (Figure 1—figure supplement 1 and Statistical Analysis) and we see a similar distribution, suggesting that no additional clones are induced over this time window. Finally, a recent paper (28) describes a complementary lineage labeling approach using a *Mesp1*-rtTA transgenic allele. Based on the timing of doxycycline administration, they define a functional window of Mesp1 activity between E6.25-E7.5, consistent with what we and others have shown.

We regret omitting a full description of our clonal analysis protocol and this is now included both in the main text as well as the Materials and Methods. In order to ensure an accurate description of clone locations, a thorough external examination was performed of each embryo. Following this, hearts were dissected and, in many instances, immunostained for labeled twin spots (GFP and Myc). Whole-mount hearts were photographed and subsequently embedded and sectioned for additional immunostaining (PECAM and tropomyosin). The use of optical clearing solutions, such as S*cale* or Clear^T^, was not necessary and thus not used.

*2) The authors make a novelty claim that they have shown that individual cardiac progenitor cells give rise to multiple cardiac cell types. This is not shown conclusively, as they do not use adequate costaining to identify cells in their*
Figure 1
*tissue sections. They should use antibodies against a myocyte marker and an endothelial marker in a 4-color stain of clonal progeny to show this*.

The reviewers are correct to point out that for technical reasons we did not initially stain for all cell-lineage markers at the same time in our tissue sections. We relied on cell morphology and staining or absence of staining for a single lineage marker (PECAM). We now have stained an additional clone example (Figure 1—figure supplement 3) that clearly shows a twin spot (Green) that gives rise to both endocardium (PECAM positive) and cardiomyocytes (Tropomyosin positive), thus providing additional confirmation that individual cardiac progenitor cells can give rise to multiple cardiac cell types.

*3) In the text and staining in*
Figure 3*. What is the evidence that Mesp1 and Smarcd3 mRNAs do not overlap? When possible, which it should be for reporters like GFP and lacZ, the authors should use fluorescence detection to colocalize markers in tissue sections. Then, arrows could be used to pinpoint cells in sections with different signatures to better make the authors' points*.

The reviewers suggest that whenever possible we should look at co-localization of *Mesp1* and *Smarcd3* in the same tissue section or embryo rather than comparing between embryos stained for each respective marker (example, Figure 3—figure supplement 1). We have now included additional images that document the non-overlapping expression of *Mesp1* mRNA and *Smarcd3*-F1 lacZ activity (Figure 3—figure supplement 1). While immunostaining for Mesp1 and Smarcd3-F1 LacZ/GFP would be the preferred method to document at the cellular level the different signatures of cells, we have not been able to identify a suitable antibody for Mesp1 immunostaining. As a result, we must rely on *Mesp1* mRNA in situ hybridization combined with immunostaining or X-gal staining for Smarcd-F1. In addition, we would like to point out that in our *in vitro* differentiation assay (Figure 6) we see a temporal distinction between the peaks of Mesp1 and Smarcd3 expression; *Mesp1* mRNA expression is mostly gone at day 4.5, the same time that *Smarcd3* mRNA expression begins to increase. These data also support the idea that at the temporal level, *Mesp1* and *Smarcd3* are non-overlapping differentiation markers.

*4) The authors use a confetti approach with a Smarcd3 enhancer driving an inducible Cre. This is a useful way to confirm the MADM results, although there may be pitfalls with tamoxifen persistence, etc. The authors mention the results briefly that labeled cells within clones are contained in single chambers, but it is better to show the full dataset here to have an idea of how they made their conclusion*.

The reviewers point out that we used a complementary approach (Confetti mice) to confirm the major conclusions from our MADM clonal analysis. We did not perform an exhaustive analysis with the Confetti mice because we felt the results from our MADM analysis were compelling and statistically significant. We focused our efforts on the MADM approach because of the advantages that we felt this system offered (rare labeling, ability to distinguish twin spots, permanent daughter cell labeling). While we attempted to perform MADM analysis with our *Smarcd3*-F1^CreERT2^ allele, the recombination frequency was too low to identify any clones in several litters of embryos. In addition, as the reviewers point out, there are pitfalls associated with tamoxifen persistence that might confound our results. Nonetheless, we crossed our *Smarcd3*-F1^CreERT2^ allele with the *ROSA*^*Confetti*^ allele and induced recombination early (E6.5) and identified 6 embryos from several litters that contained rare or infrequent labeling events and none of these examples showed a labeling pattern that would support a common progenitor contributing to both right and left ventricles. Instead, we saw clonal labeling of a single chamber. While the number of embryos we looked at is not statistically significant, we believe performing a full analysis using the *Smarcd3*-F1^CreERT2^; Confetti system in addition to the *Mesp1*^*Cre*^-MADM system is beyond the scope of this paper, but that the current data are highly supportive of our conclusions with the MADM system.

*5) The authors mention that early Tbx5+ cell labeling marks progeny corresponding to epicardial progenitors. However, they do not show cell type-specific markers to validate this colocalization. They need to do the costains or remove the statement*.

The reviewers are correct to point out that we describe an early Tbx5+ population that contributes to epicardial progenitors based on anatomic location without formally showing co-localization with an epicardial marker. We have now stained tissue sections from an early tamoxifen pulse (E6.5) harvested at E14.5 with anti-Tbx18, a marker of epicardial (and interventricular septum) cells (Zeng et al. 2011). We see rare GFP-positive/Tbx5-lineage derived cells on the epicardial surface of the heart that are also positive for Tbx18 immunostaining (Figure 7—figure supplement 1), thus confirming that an early population of Tbx5 derived cells also labels epicardial progenitors.

*[Editors' note: further revisions were requested prior to acceptance, as described below*.*]*

*Original Comment #4: The authors use a confetti approach with a Smarcd3 enhancer driving an inducible Cre. This is a useful way to confirm the MADM results, although there may be pitfalls with tamoxifen persistence, etc. The authors mention the results briefly that labeled cells within clones are contained in single chambers, but it is better to show the full dataset here to have an idea of how they made their conclusion*.

*Also, authors should indicate this small number of confetti animals examined in the text of their manuscript, and any limitation of their analysis, for the benefit of anyone who will attempt to repeat this work*.

We have added text to add the numbers of embryos studied, and emphasize the technical limitations of the use of the confetti mice. We also indicated that the MADM system has a recombination frequency too low for use with CreERT2 transgenes.

*Original Comment #5: The authors mention that early Tbx5+ cell labeling marks progeny corresponding to epicardial progenitors. However, they do not show cell type-specific markers to validate this colocalization. They need to do the costains or remove the statement*.

*Further comment in response to authors' rebuttal: Disagree. Based on the image provided, the authors should remove the statement on contribution to epicardial progenitors. The red Tbx18 signal is equivalent to background* – *the signal and claim of costain is unlikely to convince the reader*.

We agree that the data is not as convincing as it could be, and have removed the panel from Figure 7—figure supplement 1, as well as text referring to epicardial labeling.